# Sibling rivalry among the ZBTB transcription factor family: homodimers versus heterodimers

Sofia Piepoli[1,2] , Sarah Barakat[1] , Liyne Nogay[1] , Büşra Şimşek[1,2] , Umit Akkose[1] , Hakan Taskiran[1], Nazife Tolay[1] , Melike Gezen[1] , Canberk Yarkın Yeşilada[2] , Mustafa Tuncay[2] , Ogün Adebali[1] , Canan Atilgan[1] , Batu Erman[2]

The BTB domain is an oligomerization domain found in over 300 proteins encoded in the human genome. In the family of BTB domain and zinc finger–containing (ZBTB) transcription factors, 49 members share the same protein architecture. The N-terminal BTB domain is structurally conserved among the family members and serves as the dimerization site, whereas the C-terminal zinc finger motifs mediate DNA binding. The available BTB domain structures from this family reveal a natural inclination for homodimerization. In this study, we investigated the potential for heterodimer formation in the cellular environment. We selected five BTB homodimers and four heterodimer structures. We performed cell-based binding assays with fluorescent protein–BTB domain fusions to assess dimer formation. We tested the binding of several BTB pairs, and we were able to confirm the heterodimeric physical interaction between the BTB domains of PATZ1 and PATZ2, previously reported only in an interactome mapping experiment. We also found this pair to be co-expressed in several immune system cell types. Finally, we used the available structures of BTB domain dimers and newly constructed models in extended molecular dynamics simulations (500 ns) to understand the energetic determinants of homo- and heterodimer formation. We conclude that heterodimer formation, although frequently described as less preferred than homodimers, is a possible mechanism to increase the combinatorial specificity of this transcription factor family.

## Introduction

BTB (Broad complex, Tramtrack and Bric-à-brac) domains are protein–protein interaction domains that are found in over 300 human genome–encoded proteins (Letunic et al, 2021) including the N-termini of 49 zinc finger and BTB (ZBTB) proteins (Perez-Torrado et al, 2006; Siggs & Beutler, 2012). The X-ray structures of the nine members of this family that have been solved to date are from ZBTB7a (LRF) (Stogios et al, 2007), ZBTB16 (PLZF) (Ahmad et al, 1998), ZBTB17 (MIZ1) and ZBTB32 (FAZF) (Stogios et al, 2010a, 2010b), ZBTB19 (PATZ1) (Piepoli et al, 2020), ZBTB27 (BCL6) (Ahmad et al, 2003), ZBTB31 (MYNN) (Cooper et al, 2008), ZBTB33 (KAISO) (Stogios et al, 2010a, 2010b), and ZBTB48 (HKR3) (Filippakopoulos et al, 2007). These structures indicate that the BTB domain forms obligate dimers. Dimerization likely facilitates target DNA binding through the C-terminal zinc finger motifs found in the DNA-binding domains of these ZBTB transcription factors (Stogios et al, 2005). In addition to mediating homodimerization, the BTB dimer forms a scaffold for other ligands that modify the transcriptional regulation of target genes (Chevrier & Corcoran, 2014).

We recently solved the crystal structure of one ZBTB family member, PATZ1 (ZBTB19 or MAZR), from mouse and zebrafish (Piepoli et al, 2020). This work highlighted the similarity of the structures of known BTB domains. The structural similarity among the family members led us to question whether heterodimerization was possible.

Several BTB domain pairs in the ZBTB family were reported to form heterodimeric structures. These studies employ techniques that range from mass spectrometry to yeast two-hybrid screening (BioGRID database [Schmitges et al, 2016; Oughtred et al, 2021; Olivieri et al, 2021]). PATZ1, the focus of our studies, was originally identified in a two-hybrid screen with the BACH2 BTB domain used as a bait (Kobayashi et al, 2000). It is not clear that this or any other reported heterodimer has any biological function. One "forced" heterodimer X-ray structure indicates that MIZ1 and BCL6 can form stable heterodimers when expressed as a fusion protein (Stead & Wright, 2014), but whether this interaction has a physiological significance is not clear. Moreover, for many reported interactions, it is not clear that the BTB domain is sufficient for heterodimer formation, leaving the possibility that additional C-terminal residues/domains being necessary for mediating heterodimerization.

[1]Faculty of Engineering and Natural Sciences, Sabanci University, Istanbul, Turkey   [2]Department of Molecular Biology and Genetics, Faculty of Arts and Sciences, Bogazici University, Istanbul, Turkey

Correspondence: batu.erman@boun.edu.tr; ogun.adebali@sabanciuniv.edu
Sofia Piepoli's present address is Department of Microbiology and Environmental Toxicology, University of California, Santa Cruz, CA, USA.
Liyne Nogay and Hakan Taskiran's present address is Max Planck Institute of Immunobiology and Epigenetics, Freiburg im Breisgau, Germany.
Nazife Tolay's present address is Department of Biochemistry, Biocenter, University of Würzburg, Würzburg, Germany.

In the present study, we employ a fluorescent two-hybrid assay (F2H) in mammalian tissue culture cells to assess the homo-dimerization versus heterodimerization of selected BTB domains, identifying only a single pair that can stably form heterodimers. Moreover, using ImmGen data (Heng & Painter, 2008), we analyze positive and negative correlations among gene expression profiles of all ZBTB proteins in cells of the immune system. Finally, we employ molecular dynamics (MD) on a set of BTB homo- and heterodimer structures to identify if formation of homodimers or heterodimers is energetically more favorable and to determine the driving forces that contribute to dimer stability. Although one BTB domain–containing transcription factor, BACH2, contains a disulfide bond holding the obligate homodimer together in the crystal structure and in cell extracts (Rosbrook et al, 2012), our structural analysis indicates that mostly electrostatic interactions and hydrophobicity are responsible for dimer formation and stability. Among the BTB domains analyzed, only a single pair, PATZ1 and PATZ2, are co-expressed in many cell types, form heterodimers, and

have favorable binding energies. We deduce that besides the genes regulated by PATZ1-PATZ1 or PATZ2-PATZ2 homodimers, a further subset of target genes is likely regulated by PATZ1-PATZ2 heterodimers.

## Results

### Despite significant structural similarity, BTB domains prefer to form homodimers over heterodimers

The BTB domain is found in about 1% (~300) of the proteins encoded in the human genome. The core secondary structures of the BTB domain are well conserved, and their three-dimensional fold is strikingly similar (Fig 1A). BTB domains are composed of around 120 amino acids, of which 35–40% make up the conserved dimer interface. The residues forming the interface are found in secondary structure elements forming $\beta$ strands, $\alpha$ helices, and

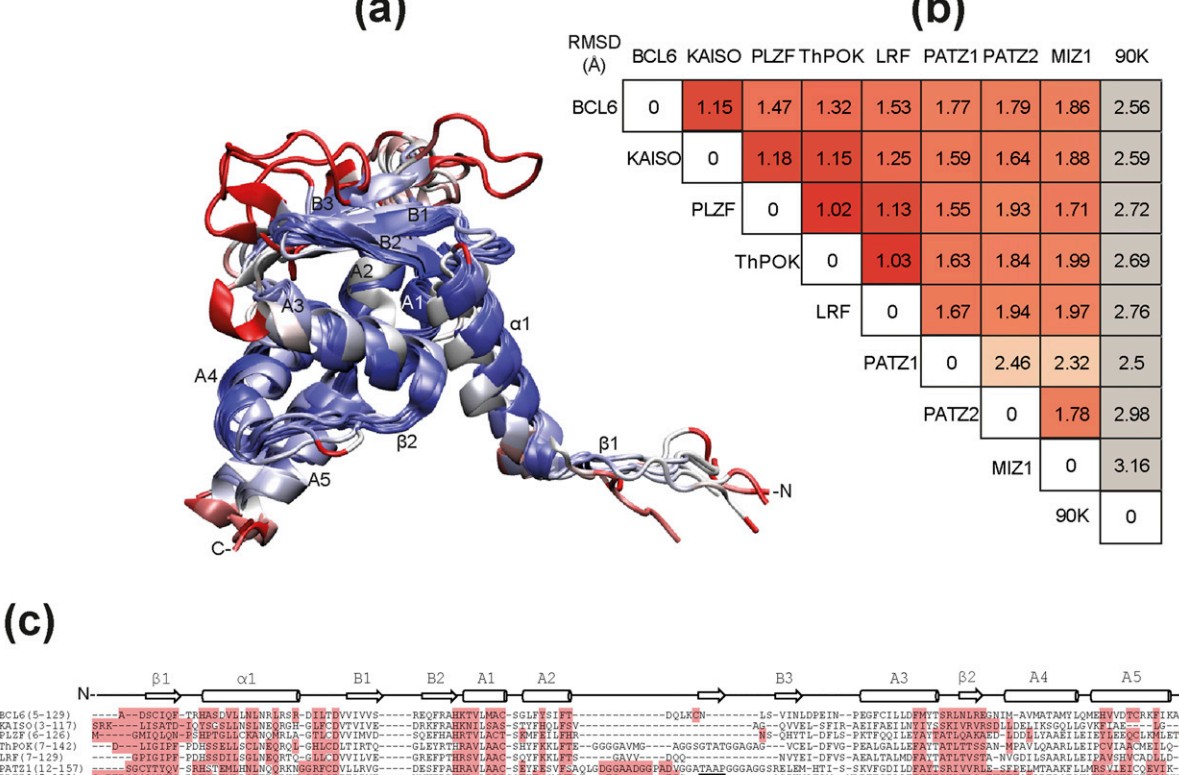

Figure 1. Structural conservation in the BTB domain.

(A) A cartoon representation of the BTB domain with annotated secondary structural elements between N- and C-termini is colored based on a metric for structural alignment (Q-score) ranging from blue to red to show the most and the least conserved regions, respectively. The nine overlapped structures belong to the BTB monomers of BCL6, KAISO, PLZF, ThPOK, LRF, PATZ1, PATZ2, MIZ1, and Galectin-3-binding protein (LG3BP/90K) human proteins. (B) The structural alignment is measured in terms of root mean square deviation (Å) of the $C_\alpha$ atoms for each pair of BTB domain structures. The root mean square deviation among this set of BTB structures is under 2 Å except for the two cases of PATZ1-PATZ2 and PATZ1-MIZ1. The secondary structure labeling follows the convention for the BTB fold as used in Stogios et al (2005). The structure and sequence of the human BTB-containing protein LG3BP/90K (PDB entry 6GFB) (Lodermeyer et al, 2018) is only used here as a divergent example to underline the similarity of the BTB domain in ZBTB proteins. (C) In the corresponding sequence alignment (C), the residues forming the BTB homodimer interface are highlighted. The residues in the BTB characteristic charged pocket are found at the beginning of B1 (negative) and between B2 and A1 (positive). The three absolutely conserved positions are indicated with an asterisk (*). The secondary structures are annotated on the sequences for orientation with part (A). The unlabeled $\beta$-strand between A2 and B3 indicates an additional secondary structure revealed in the model of PATZ1 (Piepoli et al, 2020).

loops (β1, α1, α1/B1 loop, A1, A2, A3, A3/β2 loop, β2, and A5 highlighted in Fig 1C). The presence of β1, α1, and β2 is a specific feature of the BTB domain of ZBTB proteins defined by Stogios et al (2005). Notably, the BTB domain of MIZ1 is an exception as it lacks the β1 strand. To quantify the structural similarity of BTB domains, we calculated pairwise root mean square deviation (RMSD) values for eight select ZBTB proteins, whose structures were solved, or models were easily constructed (Fig 1B). Although primary sequence conservation is only evident in subregions of the domain (Fig 1C), structural similarity ranges between 1 and 2.5 Å (Fig 1B).

To study the potential dimer formation in vitro, we setup a system to screen dimer formation of the eight aforementioned BTB domains in a pairwise fashion. We repurposed the commercially available F2H assay (ChromoTek) (Fig 2A). For this assay, each minimal BTB domain was expressed as an N-terminal fusion to either tagGFP or tagRFP fluorescent protein in the BHK-1 cell line engineered with the insertion of a large number of LacO sequences

into a genomic locus. The fusion proteins were co-expressed with a fusion protein composed of the DNA-binding domain of the LacI (lac repressor) protein fused to a GFP-binding nanobody (GBP). BHK-1 cells transiently expressing these three fusion proteins were visualized under fluorescent microscopy. A GFP focus was detected where the Lac I anchored the BTB-tagGFP fusion protein captured by the tagGFP-specific nanobody onto the locus containing the LacO sites. Association between the tagGFP- and tagRFP-tagged BTB domains also formed a co-localized red fluorescent focus indicating dimer formation. Microscopic images of the F2H assay conducted with all 64 BTB pairs were used to generate a matrix of homo- and heterodimers (Fig 2B). Of the pairs of BTB domains analyzed, we found that all could form homodimers (shown on the diagonal of the matrix and in Fig 2C and D), but only the PATZ1-PATZ2 pair formed a heterodimer in this assay (Fig 3A). We confirmed heterodimer formation between the BTB domains of PATZ1 and PATZ2 by co-immunoprecipitation in HEK-293 cells which do

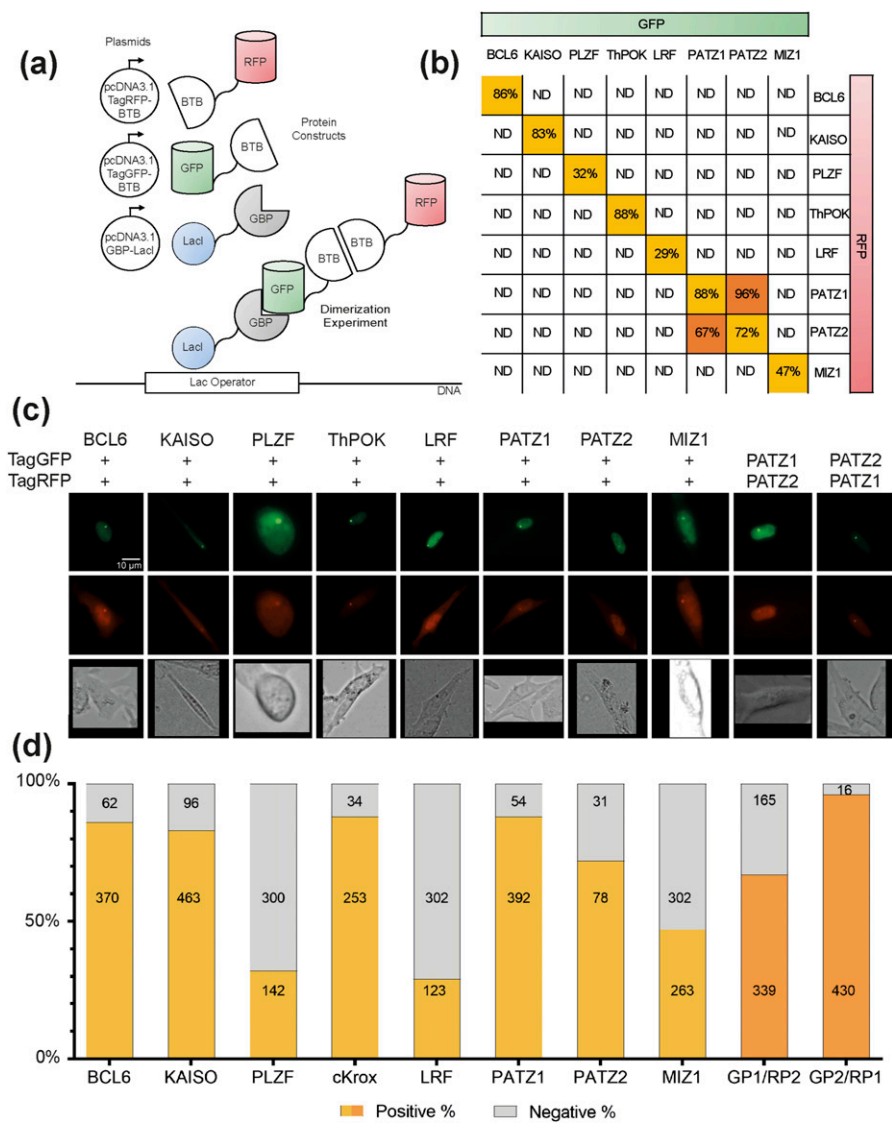

**Figure 2. F2H assay identifies BTB domain homo- and heterodimer formation in vitro.**
**(A)** Schematic description of the experimental setup. The co-transfected plasmids of the recombinant sequences of BTB domains tagged with green or red fluorescent proteins (GFP or RFP) and the GFP-binding nanobody (GBP) fused to Lac I sequence are represented as white circles next to the expressed fusion proteins. Below, a model of the interacting proteins in the co-localization experiment. **(B)** In matrix representation, the summary of interactions among the different dimer combinations. For each experimental pair, the colocalization signal is either not detected (ND) or detected in the reported percentage of the total number of cells analyzed. The only heterodimer identified with this assay is between PATZ1 and PATZ2 BTB domains. **(C)** Representative fluorescent microscopy images of colocalized tagGFP or tagRFP fusion BTB domains. Only the positive scored interactions from part (B) are shown. Three channel images displayed GFP (top row), RFP (middle row) fluorescence, and brightfield (bottom row).
**(D)** Quantification of the colocalization assay. The bar graph shows the percentage of GFP focus–positive cells that also displayed an RFP focus (positive) or not (negative). Numbers inside the bar graphs indicate the total number of cells analyzed for each case. Colors refer to part (B) where each column displays data from cells transfected with GFP- and RFP-tagged versions of the indicated BTB domains. The only heterodimers that interact were GFP-tagged PATZ1-BTB (GP1) with RFP-tagged PATZ2-BTB (RP2) and GFP-tagged PATZ2-BTB (GP2) with RFP-tagged PATZ1-BTB (RP1).

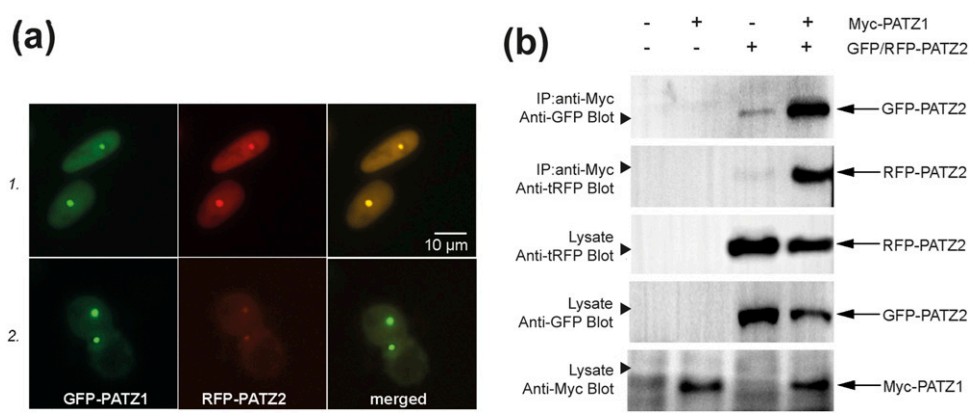

**Figure 3. PATZ1 and PATZ2 BTB domains can efficiently heterodimerize.**
**(A)** Two examples of the cells in which co-localization of the fluorescent signals from GFP-tagged PATZ1 and RFP-tagged PATZ2 BTB domains was detected. Two images of each cell were collected using different fluorescence filters, and an overlay of the two images is shown on the right. **(B)** Co-immunoprecipitation and Western blotting confirm the interaction between the BTB domains of PATZ1 and PATZ2. HEK-293 cells were transfected with the indicated constructs of Myc epitope–tagged PATZ1 and tagGFP- or tagRFP-tagged PATZ2. Anti-Myc immunoprecipitation of whole-cell lysates was followed by anti-GFP or anti-RFP Western blotting. Un-immunoprecipitated lysates are shown for loading controls. Arrowheads on the left indicate the location of size markers that correspond to 46 kD for the top four rows and to 25 kD for the bottom row.

not contain the LacO sequence in their genome and do not express the focus forming LacI-GBP protein (Fig 3B). Thus, the interaction between the BTB domains of PATZ1 and PATZ2 were independent of focus formation and the identity of the fluorescent tag used to label the proteins.

We note that in the F2H assay, not all pairs of BTB domains interact with the same efficiency (Fig 2D). For example, only 47% of the MIZ1-tagGFP foci could facilitate the formation of MIZ1-tagRFP foci (Figs 2D and S1A and B). Surprisingly, even though MIZ1-BCL6 was shown to form "forced" heterodimers (Stead & Wright, 2014), this pair did not score positively in the F2H assay, regardless of whether MIZ1-tagGFP (Fig S1C) or BCL6-tagGFP (Fig S1D) was used to recruit RFP-tagged heterodimers to the focus. We were also surprised to find that BACH2 could not heterodimerize with PATZ1 in this assay even though these proteins interact in a yeast two-hybrid assay (Kobayashi et al, 2000). Although BACH2 could readily form homodimers in this F2H assay (Fig S1E), BACH2-tagRFP could not be recruited to PATZ1-tagGFP foci (Fig S1F). Thus, MIZ1/BCL6 and BACH2/PATZ1 BTB domains are not sufficient to form heterodimers in the nuclei of live cells in this assay and likely require additional structures for heterodimerization.

To address whether the varied efficiency of homodimerization (Fig 2D) was because of steric hindrance caused by the location of the fluorescent protein in the fusion protein, we repeated these experiments with constructs that placed the tagGFP molecule to the C-terminus of the selected BTB domains (Fig S2). We find that the C-terminal localization of the fluorescent protein fusion with respect to the BTB domain does not alter dimerization preference. Crystal structures of obligate BTB homodimers demonstrate that the N-terminus of one monomer is in very close proximity to the C-terminus of the second monomer. Surprisingly, the PATZ1-PATZ2 heterodimer, which showed a strong interaction when fluorescent tags were on the N-terminus of both monomers, continued to show a strong interaction when the fluorescent protein tags were on the N- and C-termini, respectively. We conclude that the unique heterodimer between PATZ1 and PATZ2 is strong enough to withstand the presumed steric hindrance caused by placing the two fluorescent tags in close proximity.

To identify possible restrictions on heterodimer formation, we investigated the expression profiles of all ZBTB proteins in various cell types of the immune system using ImmGen data (Heng & Painter, 2008). We particularly focused on four candidate pairs of ZBTB proteins (PATZ1-PATZ2, BCL6-PATZ1, MIZ1-BCL6, and LRF-ThPOK), which were previously reported to form heterodimers (Widom et al, 2001; Stead & Wright, 2014; Vacchio et al, 2014; Huttlin et al, 2015).

Although the expression of these ZBTB genes were positively correlated in many immune system cell types, *Bcl6-Patz1* expression was negatively correlated in dendritic, mast, basophil, and eosinophil cells (Table 1 and Fig S3). The negative correlation between *Lrf-ThPOK* expression in pooled T-lymphocyte data is not evident when individual subpopulations are evaluated (Heng & Painter, 2008). In this analysis, although positive correlation does not imply physical association between ZBTB proteins, it provides evidence that the physical association between PATZ1 and PATZ2 demonstrated in the F2H assay is not restricted by expression in most immune cell types (Figs S3 and S4).

### Structurally conserved BTB domains use diverse mechanisms to stabilize homodimers

To better understand the potential of BTB domain heterodimerization, we assessed structural features that contribute to dimer stability. The interaction surface for dimerization in the ZBTB family is mostly hydrophobic and involves the N- and C-termini of the two monomers and the central α-helices and loops. This dimerization interface contains a central charged pocket that consists of two charged residues (an absolutely conserved negatively charged aspartate [D], located at the beginning of B1, and a positively charged lysine [K] or arginine [R] at the beginning of A1, which form inter- or intra-chain ionic bonds [Melnick et al, 2000]). We analyzed either crystal structures or models of five homodimers and four putative heterodimers by MD simulations to identify the relevance of these features.

For each dimer pair, we ran MD simulations of 500 ns. Our analysis of the interface interactions focused on the lifetime of salt bridges (plotted as barcode graphs) that have a strong contribution in the electrostatic component of the total ΔG of binding for five homodimers (PATZ1, BCL6, MIZ1, LRF, and PATZ2) (Fig 4) and four putative heterodimers (PATZ1-PATZ2, BCL6-PATZ1, MIZ1-BCL6, and

**Table 1. Expression correlation of four pairs of ZBTB genes[a].**

| | B cells | T cells | Monocytes | Stem cells | Stromal cells | Innate lymphocytes | Macrophages | Dendritic cells | MBE | Granulocytes |
|---|---|---|---|---|---|---|---|---|---|---|
| *Patz1-Patz2* | 0.433 | 0.229 | 0.099 | 0.354 | 0.627 | 0.245 | 0.446 | 0.331 | 0.507 | 0.581 |
| *Bcl6-Patz1* | 0.077 | **0.248** | −0.086 | −0.024 | **0.683** | 0.157 | 0.097 | **−0.361** | **−0.573** | **0.637** |
| *Bcl6-Miz1* | −0.194 | **0.356** | −0.122 | 0.176 | 0.197 | **0.603** | **0.336** | 0.046 | −0.026 | **0.598** |
| *Lrf-ThPOK* | **0.675** | **−0.185** | 0.036 | 0.334 | **0.370** | **0.455** | 0.132 | −0.080 | **0.479** | **0.499** |

[a]R-values with *P*-value ≤ 0.05 (significant) are shown in bold. The grouped mast, basophil, and eosinophil are shortened as "MBE." Additional information on these values is in Fig S3.

LRF-ThPOK) (Fig 5). Of the homodimers analyzed, we find that the PATZ1-PATZ1 pair has the highest number of interchain charged interactions (Fig 4A and Table 2). The salt bridge formed between R39 and D42 (which is in the BTB domain charged pocket) was originally observed in the crystal structure of PATZ1 (PDB entry 6GUV) but was replaced by the R39-D76 interaction upon the construction of the missing loop model. The extended MD simulation recovers the R39-D42 salt bridge. Unlike the PATZ1 homodimer, which contains dynamic salt bridges, the homodimers of BCL6, MIZ1, LRF, and PATZ2 (Fig 4B–E) have stable salt bridges forming their conserved charged pockets. Curiously, the residues of the charged pocket of BCL6 (Fig 4B) form intrachain electrostatic interactions rather than interchain bonds in the crystal structure (PDB entry 1R29) and continue to do so over the course of the simulation.

As evolutionary conservation is correlated with structural or functional roles of amino acids, we assessed the conservation score for every residue of the BTB domain. These scores are color-coded (scored from 1 to 9) in the tertiary structure of the respective BTB domains (Fig S5) and are annotated in Fig 4. We find that PATZ1 R39 and D42 are absolutely conserved oppositely charged residues. As the PATZ1 A2/B3 loop is a feature only observed in mammals, this region, including D76, shows low conservation (Fig S5 and Piepoli et al [2020]).

We surmised that the choice between homo- and heterodimer formation may be driven by the relative stability of each alternative pair. To understand the thermodynamic basis of dimerization, we calculated an estimate of the total $\Delta G$ of binding by summing the free energy of $\Delta E_{int}$, $\Delta E_{ele}$, $\Delta G_{sol}$, and $\Delta E_{vdW}$, based on MM-GBSA calculations derived from MD trajectories of homodimers (Table 2); we note that these values should only be treated as scores which are directly correlated with experimental $K_d$ values (see the Materials and Methods section for details). Calculations were restricted to the equilibrated portions of the trajectory, as shown by the boxed portions of the RMSD plots in Fig 4. As expected from the stable homodimeric structure of BTB domains, the energy features contributing to the dimerization interface for all dimers resulted in energetically favorable interactions with negative $\Delta G$ values. We find that although the stabilization energy per residue varied in the interval [–2.1, –1.6] kcal/mol, the factors contributing to this energy were from different sources for each pair of homodimers. For the intramolecular interactions in the molecules making up the dimer, for all systems analyzed, the bond stretching/bending/torsions ($\Delta E_{int}$) which make up the local terms were all negative, indicating that local strains were relieved upon dimerization, more so in some systems (e.g., LRF homodimer) than in others (e.g., BCL6 or

MIZ1 homodimer). In terms of nonbonded interactions, we found that the PATZ1 homodimer is overwhelmingly stabilized by the large favorable electrostatic interactions ($\Delta E_{ele}$), especially those established at the interface as is also corroborated by the salt bridges formed (Fig 4A).

MM-GBSA calculations show that the PATZ1 BTB domain is the most favorable homodimer with binding free energy ($\Delta G$) equal to –529.1 kcal/mol for the equilibrated conformation, averaged between two duplicate MD runs (Table 2). BCL6 BTB homodimer is a less favorable construct than that of PATZ1, having 0.2 kcal/mol higher binding free energy per amino acid (–1.8 versus –1.6 kcal/mol). In this homodimer, the energy component deriving from local constraints in bonds, angles, and dihedrals ($\Delta E_{int}$) is the least favorable. Because of low variation in the RMSD (Fig 4B), for BCL6, the whole trajectory was considered for MM-GBSA calculations. In the case of the MIZ1 BTB homodimer, we considered the equilibrated portion between 100 and 500 ns (Fig 4C and Table 2). The binding free energy is favorable and equal to –384.1 kcal/mol, yet along with the previous BCL6 case, is the least favorable among the dimers analyzed in this study (–1.6 kcal/mol/AA).

Despite the similar binding energies, the factors contributing to the overall energy are different. The energy components contributing favorably to a loss of electrostatics are the van der Waals energy ($\Delta E_{vdW}$) and the solvation free energy, deriving from the nonpolar contribution ($\Delta G_{sol}^{SA}$) making a weak dimer interface for MIZ1 BTB protein. In comparison, the binding free energy for LRF homodimer is the most favorable on a per amino acid basis (–2.1 kcal/mol/AA). Unlike in the PATZ1 homodimer, this strength draws not from an abundance of electrostatic interactions at the interface but rather is because of the local release of strains in bond stretching, bending, and torsional angles that occurs upon binding.

Apart from the formation of salt bridges and energetic contributions, another factor influencing the choice between homo- and heterodimer may be the surface area of a monomer buried by dimerization; this quantity contributes strongly to the $\Delta G_{sol}^{SA}$ term in Equation (1). Thus, we extracted the solvent accessible surface area (SASA) of the dimers as well as their monomeric forms and calculated the resultant buried surface area (BSA) from the trajectories of the five BTB homodimers (Fig S6 and Table 2). We find that PATZ1 and PATZ2 have the largest BSA, correlating with the largest calculated free energy change of homodimerization (Table 2). The variability of the BSA values over the course of the simulation shows the stability of all the interchain contacts, including ionic, polar, and nonpolar interactions. We therefore conclude that

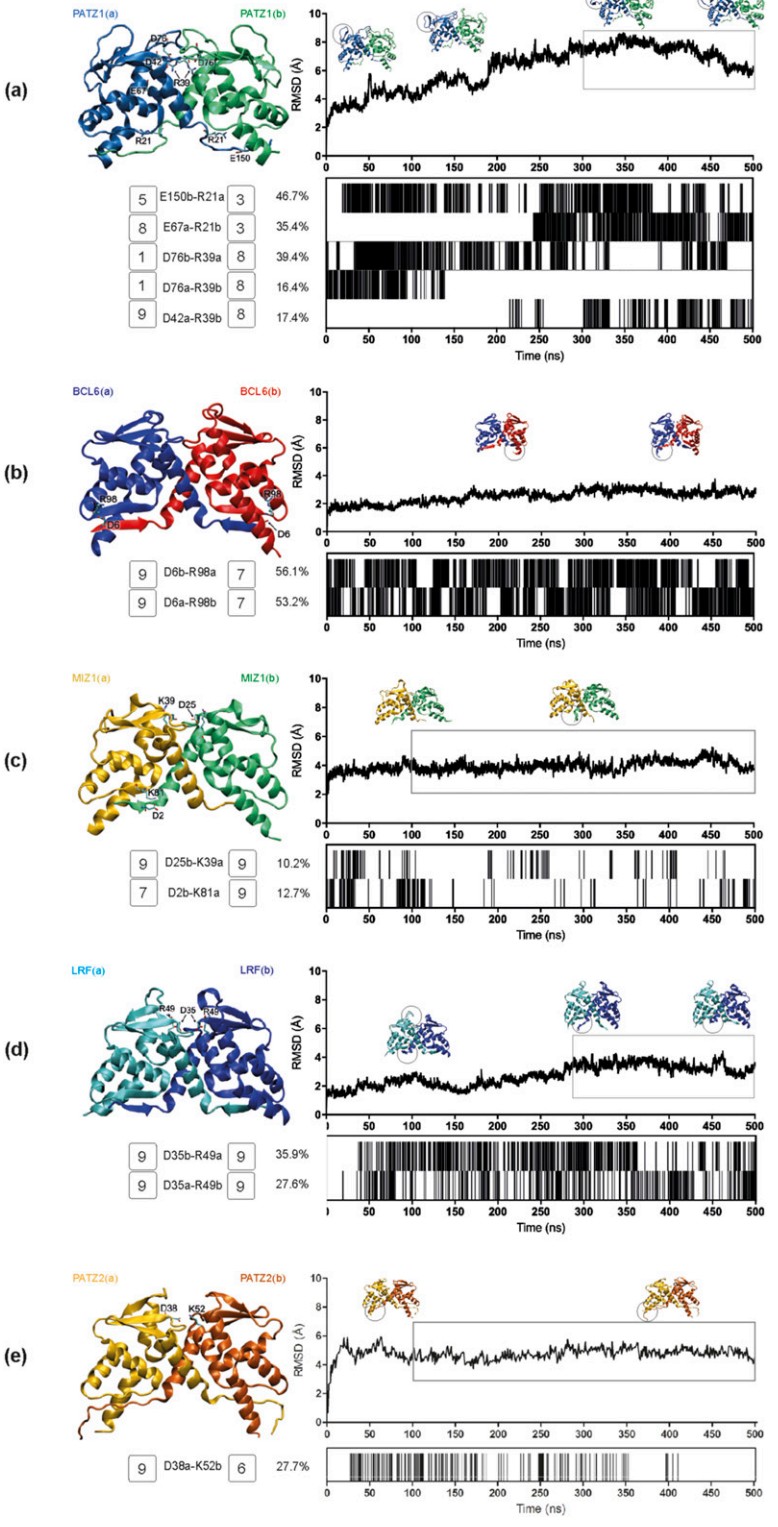

**Figure 4. Molecular dynamic simulation analyses for the BTB domain homodimers.**
**(A, B, C, D, E)** Root mean square deviation plots, salt bridge formation barcodes, and a cartoon representation of the BTB domain dimer structure are shown for, PATZ1 (A), BCL6 (B), MIZ1 (C), LRF (D), and PATZ2 (E). The root mean square deviation plot shows the structural distance (Å) of the protein atoms coordinates ($C_\alpha$) as a function of time (ns) and contains the snapshots of the significant conformational changes of the dimer structure. Every salt bridge between a pair of charged amino acids with a distance within the 3.0-Å cutoff is represented with a bar in the barcode plot and reported if present over the 8% of the total simulation time. The amino acids belonging to one monomer (a) or the other (b) involved in the interchain interactions are labeled with one-letter code. For each residue in these interchain salt bridges, the conservation score is displayed next to its label in the range [1,9], increasing from variable (1) to conserved (9) as calculated via the ConSurf web server.

although the overall folds of the BTB domains are well conserved as implicated by the low RMSD values (Fig 1B), energetically, dimerization is not facilitated by a single mechanism. In fact, it is predominantly the extensive salt bridge formation in PATZ1, release of local strains in LRF, the relatively low energy cost of electrostatic solvation for MIZ1, and hydrophobicity for PATZ2. For BCL6, it is a combination and compensation of all these factors that achieve the final homodimer.

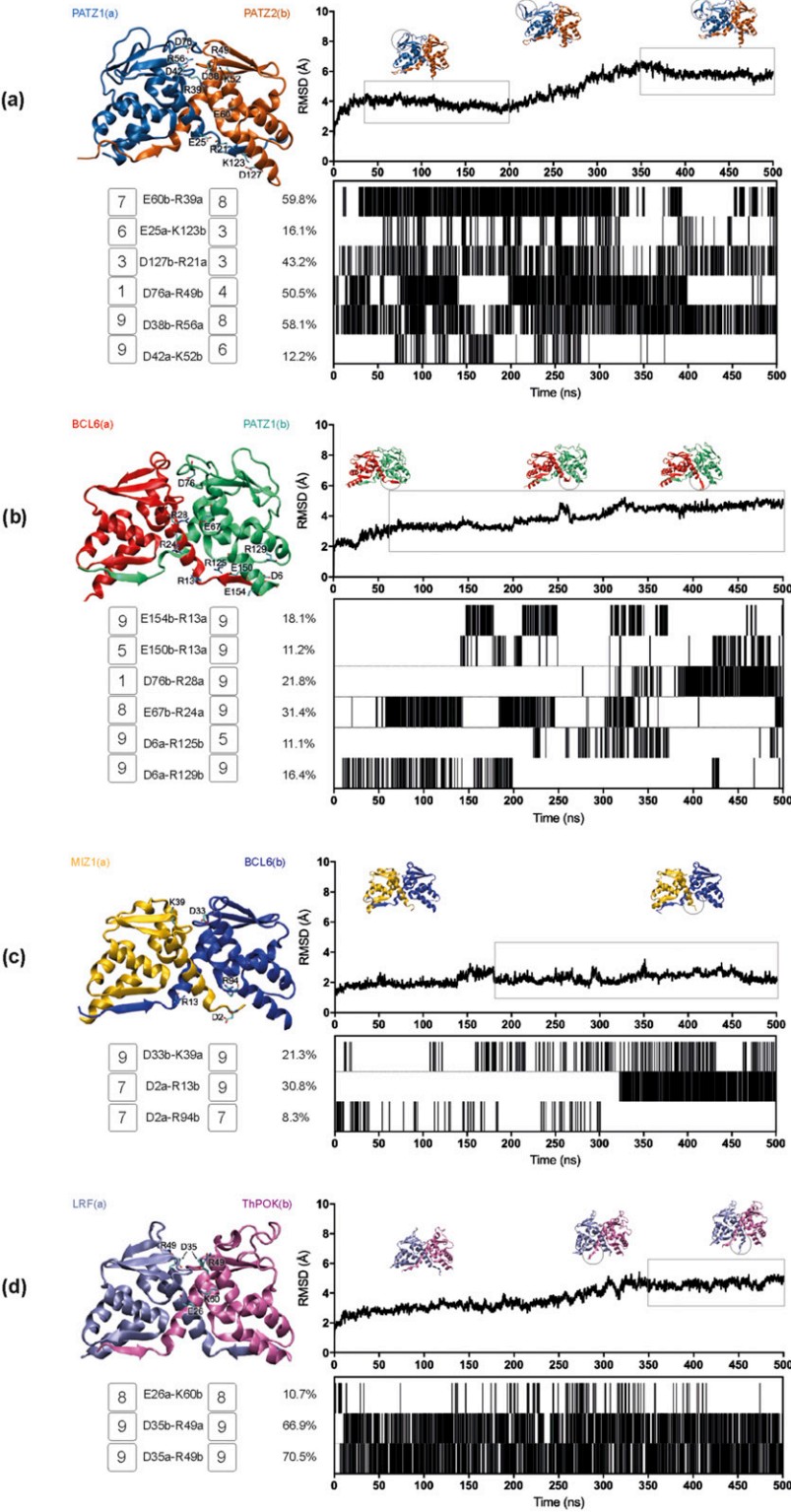

**Figure 5. Molecular dynamic simulation analyses for the BTB domain heterodimers.**
**(A, B, C, D)** Root mean square deviation plots, salt bridge formation barcodes, and a cartoon representation of the BTB domain dimer structure are shown for PATZ1-PATZ2 (A), BCL6-PATZ1 (B), MIZ1-BCL6 (C), and LRF-ThPOK (D). See caption to Fig 4 for details.

## Heterodimerization results from additional interactions

To determine the relative stability of heterodimers, compared with homodimers, we constructed models of four putative BTB heterodimers originating from monomer structures and ran MD simulations, performed MM-GBSA calculations and identified the BSA values. As before, for each heterodimer pair, MD simulations (500 ns) yielded information about interface interactions based on the lifetime of salt bridges (Fig 5 and Table 3).

**Table 2. Binding energies for BTB homodimers and individual contributions to the total energy[a].**

| Homodimers | MM-GBSA | | | $\Delta G_{sol}$ | | | | | | | |
| | $\Delta G$ binding kcal/mol | $\Delta E_{int}$ | $\Delta E_{ele}$ | $\Delta G_{sol}^{PB}$ | $\Delta G_{sol}^{SA}$ | $\Delta E_{vdW}$ | #AA dimer | $\Delta G$/ AA | avg BSA (Å²) | AA in interface (%) | Avg #salt bridges at interface |
|---|---|---|---|---|---|---|---|---|---|---|---|
| PATZ1 (300–500 ns) | **−529.1 ± 0.7** | −353.9 ± 0.4 | **−418.0 ± 3.5** | 437.1 ± 3.2 | 0 ± 0.1 | −194.3 ± 0.5 | 290 | −1.8 | 1,864 ± 14 | 40.7 | 1.7 ± 0.1 |
| BCL6 (1–500 ns) | −401.6 ± 0.2 | −237.5 ± 0.1 | −278.7 ± 0.9 | 330.6 ± 0.8 | −25.4 ± 0.1 | −190.5 ± 0.1 | 250 | −1.6 | 1,899 ± 6 | 34.8 | 1.2 ± 0.1 |
| MIZ1 (100–500 ns) | −384.1 ± 0.2 | −282.8 ± 0.1 | −170.0 ± 1.0 | **235.9 ± 1.0** | −18.8 ± 0.1 | −148.4 ± 0.2 | 234 | −1.6 | 1,471 ± 4 | 37.2 | 0.3 ± 0.1 |
| LRF (280–500 ns) | −520.3 ± 0.3 | **−380.0 ± 0.2** | −322.8 ± 1.5 | 399.6 ± 1.4 | −22.3 ± 0.1 | −194.7 ± 0.2 | 244 | **−2.1** | 1,752 ± 4 | 36.9 | 0.8 ± 0.1 |
| PATZ2 (100–500 ns) | −485.4 ± 0.7 | −335.5 ± 0.3 | −247.8 ± 3.9 | 254.5 ± 3.8 | **43.3 ± 0.1** | **−200.0 ± 0.6** | 258 | −1.9 | 2,037 ± 6 | 38.0 | 0.3 ± 0.1 |

[a]Calculations are carried out for the equilibrated portion of the trajectory indicated in parenthesis and shown in Fig 4. Most favorable energy values indicated in bold. SEs are indicated for each energy term.

**Table 3. Binding energies for BTB heterodimers and individual contributions to the total energy[a].**

| Heterodimers | $\Delta G$ binding kcal/mol (expected) | MM-GBSA | | | $\Delta G_{sol}$ | | | | | | | |
| | | $\Delta G$ binding kcal/mol | $\Delta E_{int}$ | $\Delta E_{ele}$ | $\Delta G_{sol}^{PB}$ | $\Delta G_{sol}^{SA}$ | $\Delta E_{vdW}$ | #AA dimer | $\Delta G$/ AA | avg BSA (Å²) | AA in interface (%) | Avg #salt bridges at interface |
|---|---|---|---|---|---|---|---|---|---|---|---|---|
| PATZ1-PATZ2 (350–500 ns) | −507.3 ± 1.4 | **−529.5 ± 0.4** | −340.0 ± 0.2 | **−471.0 ± 2.8** | 535.4 ± 2.7 | **−30.4 ± 0.1** | **−223.4 ± 0.3** | 274 | **−1.9** | 2,348 ± 5 | 40.1 | 2.5 ± 0.1 |
| PATZ1-BCL6 (60–500 ns) | −465.4 ± 0.9 | −470.9 ± 0.2 | −306.6 ± 0.1 | −461.6 ± 1.8 | 531.0 ± 1.7 | −27.3 ± 0.1 | −206.4 ± 0.2 | 270 | −1.7 | 2,054 ± 6 | 37.8 | 1.3 ± 0.1 |
| MIZ1-BCL6 (180–500 ns) | −392.9 ± 0.4 | −397.6 ± 0.2 | −272.3 ± 0.1 | −152.8 ± 0.8 | **202.7 ± 0.7** | −19.6 ± 0.1 | −155.7 ± 0.2 | 242 | −1.6 | 1,548 ± 4 | 33.1 | 0.6 ± 0.1 |
| LRF-ThPOK (350–500 ns) | | −471.3 ± 0.3 | **−348.6 ± 0.2** | −249.3 ± 1.5 | 316.4 ± 1.4 | −21.0 ± 0.1 | −168.8 ± 0.2 | 259 | −1.8 | 1,706 ± 4 | 34.7 | 1.5 ± 0.1 |

[a]Calculations are carried out for the equilibrated portion of the trajectory indicated in parenthesis and shown in Fig 5. Most favorable energy values indicated in bold. SEs are indicated for each energy term.

Although the PATZ1 interface has the largest number of salt bridges among the homodimers, the PATZ1-PATZ2 and BCL6-PATZ1 heterodimer interfaces established additional salt bridges (Fig 5A and B). The PATZ1-PATZ2 dimer interface has a significant inter-action between residues E60b and R39a, both well-conserved residues, which is present for 60% of the trajectory. A second salt bridge formed between D38b and R56a, also a well-conserved pair, is present for over 58% of the trajectory. Interestingly, although all other PATZ1 residues involved in the salt bridges between PATZ1-PATZ2 heterodimers also make similar interactions in the PATZ1 homodimer, the PATZ1 residue R56 only makes salt bridges with PATZ2 (reconstituting an interchain charged pocket interaction). In general, the residues forming the charged pocket in the hetero-dimer models form interchain salt bridges, with the exception of the BCL6-PATZ1 pair, which retains stable intrachain salt bridges.

BTB domain N-terminal interactions have recently been pro-posed to mediate dimer stability (Mena et al, 2020). N-terminal β-strand spontaneous complex dissociation could thus differen-tiate homodimers from heterodimers. Although we observe the presence of stable N-terminal β-strand interactions in many of the homodimer structures, this feature is present only in the PATZ1-PATZ2 heterodimer model (Fig 5A). Noticeably, the two symmetrical β-sheets formed at the dimerization interface between β1 and β2 of the two monomers are stable throughout the PATZ1-PATZ2 simu-lation and do not show any sign of spontaneous unfolding. In contrast, in the BCL6-PATZ1 BTB heterodimer (Fig 5B), the N-ter-minal β1 strand of BCL6 (chain a) disengages from the β-sheet with β2 of PATZ1, leading to a partial unfolding of the dimer interface.

Also, the MIZ1-BCL6 heterodimer has an unstable interface because, although the BCL6 homodimer interface relies on a sheet formed by the interaction of β1 and β2, MIZ1 lacks a complete β1-strand (Fig 5C). MD trajectories reveal the accommodation of a new stable conformation for the short N-terminus of MIZ1 that swings from the initial docked position parallel to β2 in BCL6 to a new interaction with the N-terminal of BCL6. Explicitly, we can follow this conformational change by tracking the salt bridges formed by D2a initially interacting with R94b and then settling for R13b. Signifi-cantly, the spontaneous unfolding of one of the primary dimer

interface β-sheets may represent a target for dimer quality-control mechanisms (Mena et al, 2020). As for the MIZ1-BCL6 heterodimer, besides this local flexibility, the ionic interaction between the highly conserved charged pocket residues (D33b-K39a) is preserved and remains important in the trajectory of all dimers (Fig 5C). The equilibrated conformation of MIZ1-BCL6 resulting from our MD simulation is comparable (RMSD < 1.5 Å) to the deposited crystal structure for the MIZ1-BCL6 BTB domain (PDB entry 4U2M) (Stead & Wright, 2014). Similarly, we can see the highly conserved charged pocket residues D35 and R49 of the LRF-ThPOK BTB heterodimer (Fig 5D), forming two strong symmetric salt bridges both present for at least 70% of the whole trajectory.

The A2/B3 loop of PATZ1 contributes to the large BSA of the PATZ1-PATZ2 and BCL6-PATZ1 heterodimers, which have an average area of 2,348 and 2,054 Å$^2$, respectively (Fig S7). Unlike the first two cases, the MIZ1-BCL6 BTB heterodimer interface area is small, equal on average to 1,548 Å$^2$ with the lowest percentage of the total residue count involved in the interface (Table 3). This is because of an asymmetric dimer interface between the two monomers. The fluctuations in the RMSD (Fig 5C) reflect the adjustments related to the shorter β1 sequence of MIZ1-BTB.

As for the homodimers, all heterodimers show favorable interaction energy (Table 3). PATZ1-PATZ2 is the strongest heterodimer among the ones analyzed with binding free energy (ΔG) equal to −529.5 kcal/mol. BCL6-PATZ1 heterodimer is also a favorable construct with binding free energy equal to −470.9 kcal/mol. The MIZ1-BCL6 heterodimer is the least favorable of the heterodimers considered in this study with binding free energy equal to −1.6 kcal/mol/AA. The LRF-ThPOK heterodimer is a favorable construct with binding free energy of −1.8 kcal/mol/AA on the order similar to that of the PATZ1 homodimer. At the outset, a heterodimer is expected to form if the energy gain is lower than that expected from its homodimers. For example, for the PATZ1-PATZ2 heterodimer, an expected energy is the average from their homodimers, therefore, ca. −507 kcal/mol. We find that, the ΔG for the PATZ1-PATZ2 heterodimer is −529.5 kcal/mol, therefore ~22 kcal/mol lower than the average energy expected from the homodimers (Table 3). This is in contrast with the observations for BCL6-PATZ1 and MIZ1-BCL6 heterodimers, whereby the expected and measured ΔG values are within ~5–6 kcal/mol of each other; therefore, there is no substantial need to prefer heterodimers over homodimers for these pairs.

## Discussion

This study documents that BTB domains can heterodimerize. We evaluated the dimerization potential of 64 pairs of BTB domains and find that although all pairs can generate homodimers, only one, PATZ1 (ZBTB19) and PATZ2 (ZBTB24), can form heterodimers in vivo. Energetic calculations confirmed that this heterodimer could form a favorable interaction interface, predominantly because of additional stable salt bridges. Despite the similar name, PATZ1 and PATZ2 only show 26.5% identity and 42.4% similarity in their BTB domain sequence. These two ZBTB family members are structurally related, being the only proteins in the ZBTB family that have an additional AT-hook motif (binding the minor groove of adenine–thymine–rich DNA) that is thought to confer an alternative DNA-binding specificity to these proteins. In our assays, we used the minimal BTB domain consisting of 157 amino acids for PATZ1 and 133 for PATZ2, lacking the AT-hook motif. This demonstrates that the AT-hook is not necessary for heterodimer formation and that BTB domains are sufficient to form heterodimeric structures. These findings reveal that the PATZ1-PATZ2 heterodimer is as stable as the PATZ1 or PATZ2 homodimers in the cellular environment, a finding that is supported by the calculated binding free energy of these complexes. Electrostatic interactions in proteins are fine-tuned by the various niches in the cellular environment, with differences of pH or ionic strength (Sensoy et al, 2017). The dominance of the electrostatic component in the PATZ1-PATZ2 heterodimer might confer its ubiquity in the different cell types where they are co-expressed (Table 1).

The demonstration of definitive heterodimer formation between PATZ1 and PATZ2 now will allow the questioning of the participation of each protein in the phenotypes observed in the mutation or knockout of the other factor. For example, mutations in the Zbtb24 gene result in the methylation defects observed in the immuno-deficiency, centromeric instability, and facial defect syndrome type 2 (ICF2) (de Greef et al, 2011; Wu et al, 2016; Thompson et al, 2018). Does PATZ1 participate in this defect? How many of the previously identified 187 differentially expressed genes in Patz1−/− cells (Keskin et al, 2015) are controlled by PATZ1 in collaboration with PATZ2 is an open question.

We investigated the underlying structural factors behind BTB domain dimerization to understand the basis of the homodimer versus heterodimer choice. A functional consequence of homodimer formation in various ZBTB proteins is the formation of a lateral groove that is a docking site for co-repressor proteins (Melnick et al, 2002). Although other BTB domains have been shown to interact with co-repressors, the only available co-crystal structure is that of BCL6 and its co-repressors (Ahmad et al, 2003; Bilic et al, 2006; Ghetu et al, 2008; Zacharchenko & Wright, 2021). In these structures, the co-repressor peptides associate with the BTB homodimer as symmetrical pairs themselves. The interaction of BCL6 homodimers with co-repressor peptides has been studied in detail using MD supported by MM-GBSA calculations, revealing potential sites that can be targeted by drugs (Granadino-Roldan et al, 2014). With the definitive demonstration of the presence of heterodimers, we open the question of whether heterodimers can also form the landing pad structures for these co-repressors. If so, could the non-symmetrical lateral grooves of BTB heterodimers provide a mechanism of altered specificity for co-repressors? Besides the BTB domain lateral groove interactions assisted by lower β-sheet extensions, exemplified by the BCOR/NCOR1/NCOR2 interactions with BCL6, a novel interaction site on BTB domains was recently revealed (Orth et al, 2021). The interaction of a β-strand containing peptide from HUWE1 with the flexible B3 region of MIZ1 can result in an upper β-sheet extension. Whether these interactions can form in other BTB pairs is not known. An obvious candidate for such an interaction would be the flexible top region containing BTB domains such as PATZ1 and PATZ1 containing (heterodimeric) complexes.

Formation of BTB heterodimers would dramatically increase the combinatorial target specificity of this transcription factor family.

Obviously, such heterodimer formation would be restricted by the tissue- and stage-specific expression of the individual proteins. We investigated the co-expression between the protein pairs of interest, and identified a cluster of 22 co-expressed ZBTB family genes (Fig S4, left bottom corner). Co-expression likely reflects co-regulation which can also be interpreted as a prerequisite for heterodimeric interaction in multiple immune lineages. Mechanistic constraints in the synthesis of these proteins, such as the recently reported co-translational dimerization pathways (Bertolini et al, 2021), may impart restrictions on the formation of heterodimers, possibly favoring the formation of homodimers co-translated on polysomes. However, the combinatorial specificity may not be regulated only at the level of the formation of homo- or heterodimers but also in the cellular half-life of these alternative protein structures. A recent study proposed the presence of evolutionarily conserved degron residues which preferentially target BTB heterodimers for degradation (Mena et al, 2018). Although this study examined the degradation properties of non-transcription factor BTB domain containing proteins, degron structures may likely be conserved in ZBTB proteins as well, making unwanted BTB heterodimers prone to degradation. Furthermore, according to the BTB quality control hypothesis (Mena et al, 2020), heterodimers can be targeted for degradation based on the identity of the N-terminal $\beta$1 sequence that forms a critical interface surface. In fact, we identified an N-terminal sequence in the PATZ1 crystal structure that preferentially stabilizes homodimeric structures (Piepoli et al, 2020). The propensity of this region to result in aggregation that potentially targets BTB domains for degradation has also been observed in the BCL6 protein crystal structure, which can be used as a means for co-crystallization (Zacharchenko & Wright, 2021).

The F2H assay we introduce in this study is built on a previous iteration that tested the interaction between the minimal interaction domains of the p53 and MDM2/MDM4 proteins (Zolghadr et al, 2008). This system can be used as a high-throughput screening tool to test for drugs that block interaction (Yurlova et al, 2014). In its current version, this assay can be used to not only discover new heterodimers and their third-party interactors but also inhibitors of dimers. As BTB domains form obligate homodimers, it is surprising that heterodimers can in fact be observed in this assay. Because the system is set up with one monomer (tagGFP partner) with a nuclear localization signal (NLS) and a second monomer (tagRFP partner) without any such signal, we find that the interaction between BTB monomers is strong enough to recruit BTB domains with no NLS into the nucleus. Significantly, the PATZ1-PATZ2 interaction that scores positive with a GFP-RFP pair also does so with an RFP-GFP pair, indicating the robustness of the system to recapitulate in vivo interactions (Fig 2).

In this study, we determined the driving forces that contribute to dimer stability. We find by MD simulations that all heterodimers are favorable. Different mechanisms contribute to homo- and heterodimer stability. Significantly, homo- and heterodimer interfaces are typically characterized by numerous and sometimes short-lived electrostatic interactions. Thus, evolution has favored conserving the fold which serves as a template for catering to the overall functions attributed to these systems while diverse mechanisms have been used to compensate for the variations observed in family members (siblings) introduced to enable those functions. The

analysis of the energy components contributing to dimerization also paves the way to design stable BTB heterodimers particularly by engineering interface residues and limiting accessibility to degron positions. Our analysis confirms that heterodimerization among ZBTB family members is infrequent and that homodimers are preferred. Nevertheless, the absence of energetic restrictions for BTB domain-mediated heterodimers suggest that more pairs of heterodimers could possibly form, increasing transcription factor combinatorial specificity.

# Materials and Methods

### BTB domain and GBP-LacI cloning

The coding sequence of the BTB domain of selected ZBTB family proteins was amplified from cDNA derived from the human HCT116 cell line using Q5 High-Fidelity DNA Polymerase (NEB) (Table S1). Specifically for the PATZ1 expression construct, the BTB domain was amplified from a murine *Patz1* cDNA. The murine and human proteins differ at a single position (residue 91, within the A2/B3 loop) which is Ala or Thr, respectively. Amplified fragments (Table S1) were cloned into the pcDNA™3.1/*Myc*-His(–)Bexpression vector that contained either a TagGFP cDNA with a NLS or a TagRFP cDNA with no signal. BTB cDNAs were cloned into the XhoI and NotI restriction sites for TagGFP and between SmaI and NotI for TagRFP vectors, such that they encoded NLS-tagGFP-BTB or tagRFP-BTB proteins. For C-terminal FP fusion constructs, amplified fragments encoding BTB domains with primers containing NheI and HindIII were cloned into the appropriate sites of either pAC-TagRFP (*Miz1*, *Plzf*) or pLC-TagGFP (*Patz1*, *Miz1*, *Plzf*, *Lrf*) plasmids (ChromoTek), from which nanobody sequences were removed, such that they encoded BTB-tagGFP or BTB-tagRFP proteins. The recombinant plasmid DNA was sequenced and transfected into Baby Hamster Kidney fibroblasts (BHK-1 cells) that were modified to contain concatemeric *Escherichia coli* lactose operator (Lac O) sequences inserted into a single locus (ChromoTek).

For targeting the GFP fusion protein to the Lac O locus, we constructed a plasmid containing the Lac repressor sequence (Lac I) fused to a nanobody specific to GFP (GFP binding protein-GBP), derived from the *Camelus dromedarius* VHH domain cAbGFP4 (PDB structure reference: 3OGO [Kubala et al, 2010]).This fusion gene was amplified and cloned into the pcDNA™3.1/*myc*-His(–)B expression vector using NheI- and BamHI-digested amplicons generated from the F2H platform mixture as a template with forward and reverse oligonucleotides (Table S1). This plasmid encodes a fusion protein that has a 107–amino acid N-terminal GBP fused to a 355–amino acid C-terminal Lac I domain separated by a 7–amino acid linker. The experimental approach for using the LacI-GFP nanobody (GBP) to recruit GFP-tagged proteins to the LacO locus is well characterized (Herce et al, 2013; Tang et al, 2013).

### Western blotting

Whole-cell lysates (10 mM Hepes-KOH [pH 7.9], 2 mM MgCl$_2$, 0.1 mMEDTA, 10 mM KCl, 0.5% NP-40) of HEK293T cells transfected with

plasmids encoding Myc-PATZ1 (Keskin et al, 2015) and tagGFP- or tagRFP-PATZ2 were immunoprecipitated with anti–c-Myc magnetic beads (88842; Thermo Fisher Scientific). Bound beads were washed five times with ice-cold wash buffer (50 mM Hepes-KOH [pH 7.9], 100 mM KCl, 2% NP-40) and boiled in 1X Laemmli Buffer for 5 min. Precipitated proteins were resolved on 14% SDS–PAGE gels, transferred onto polyvinylidene difluoride (PVDF) membranes (88518; Thermo Fisher Scientific), and blotted with peroxidase-coupled anti-Myc (11814150001; Roche) or anti-GFP or anti-tRFP antibodies (Evrogen AB011, AB233), followed by anti-rabbit IgG–peroxidase (7074; Cell Signaling). Reactivity was revealed by enhanced chemiluminescence (ECL) (34580; Thermo Fisher Scientific) and visualized using a G-BOX Chemi XX6 documentation system (Syngene).

### Transfection, live-cell microscopy, and F2H assay

The fluorescent two-hybrid (F2H) assay (Rothbauer et al, 2006; Zolghadr et al, 2008, 2012) was used to study dimer formation between pairs of BTB domains. $1.5 \times 10^5$ BHK-1 cells were seeded into six-well plates with coverslip bottoms and transfected with polyethyleneimine (PEI) reagent at a ratio of 1:3 (DNA:PEI wt/wt). Equal mixtures of NLS-tagGFP-BTB or tagRFP-BTB and GBP-Lac I encoding plasmids were transiently co-transfected. 24 h after transfection, adherent cells were visualized using an invert fluorescent microscope (ZEISS Axio Observer Z1) with 10–20× magnification. Excitation was performed using either an HXP 120V fluorescent light source or a Colibri7 light source with LED470 or LED-Neutralwhite (540–580 nm) and Filterset 38 (Excitation 470/40 BP; dichroic 495LP; emission 525/50 BP) or Filterset 43 (Excitation 545/25 BP; dichroic 570LP; emission 605/70 BP) for tagGFP and for tagRFP visualization, respectively. Emission was detected either using a Zeiss Axiocam 503 mono or MRc5 camera.

In the F2H assay, GFP foci were only evident when the GBP-Lac I– and tagGFP-BTB–encoding plasmids were included in the transfection mixture. GFP-RFP colocalization was only evident when GBP-Lac I–, tagGFP-BTB–, and tagRFP-BTB–encoding plasmids were included in the transfection mixture. No foci were observed if the GBP-Lac I–encoding plasmid was omitted from the transfection mixture. Because the F2H-BHK cells were not synchronized in their cell cycle, some cells were in the S-phase and contained two tagGFP foci, resulting from duplicated chromosomes. For these instances, both foci were scored as independent events. Colocalization analysis was performed manually or by using the JACoP plugin of the Fiji software version 2.1.0/1.53c (quantified in Tables S2 and S3) (Bolte & Cordelières, 2006; Schneider et al, 2012; Schindelin et al, 2012). Colocalization was defined as the percentage of GFP foci–positive cells that were also positive for RFP foci (Table S2). We note that most of the GFP expressing cells were positive for GFP foci (Table S3).

### ImmGen cell type analysis of RNA co-expression

The gene expression data of 46 of the 49 ZBTB family genes were obtained from the Immunological Genome Project (ImmGen) Microarray Phase 1 and Phase 2 datasets (Heng & Painter, 2008). Probes for ZBTB21 (ZNF295), ZBTB35 (ZNF131), and ZBTB47 (ZNF651)

were missing in the dataset and were not analyzed. The dataset contained gene expression data from primary murine cells from multiple immune lineages including B lymphocytes; monocytes; mast, basophil, and eosinophil (MBE); stromal cells; innate lymphocytes; granulocytes; macrophages; dendritic cells; stem cells; and T lymphocytes. Correlation coefficients of all pairs were calculated using least-squares linear regression, and two-sided *P*-value was used for hypothesis testing.

### Conservation analysis

To retrieve homologs for each of the six BTB-domain proteins (Fig S4), Blast (Altschul et al, 1997) search was conducted locally against a nonredundant database (downloaded from Uniprot [Bateman et al, 2019] August 2019 release) including a canonical isoform for each protein. MAFFT (Katoh et al, 2002) was used to build multiple sequence alignment (MSA). We reconstructed a phylogenetic tree for each protein separately with FastTree (Price et al, 2010). We selected orthologous protein sequences from each tree, by traversing the phylogenetic tree starting from the query sequence until the node having the next human protein sequence as an eventual child. The previous node was selected as the monophyletic clade including the orthologous sequences only. Then, a new MSA and a new phylogenetic tree were built using the orthologs. The MSA is constructed with 101 orthologous sequences for PATZ1, 118 for BCL6, 88 for MIZ1, 76 for LRF, 152 for PATZ2, and 75 for ThPOK. ConSurf web server (Ashkenazy et al, 2016) was used with the final MSA and phylogenetic tree as inputs to calculate the conservation scores of the positions. Finally RAxML-NG (Kozlov et al, 2019) was used for building the phylogenetic tree of 49 ZBTB proteins, by using top 10 blast hits for each protein (Fig S8).

### Structure of heterodimers: docking and modeling

The four BTB heterodimer structures presented in this work were obtained from available crystal structures or newly modeled structures built by homology and docked monomers of homodimer structures. Among the BTB heterodimers between ZBTB family members, the MIZ1-BCL6 construct is currently the only one for which the crystal structure has been deposited (Stead & Wright, 2014) (PDB entry 4U2M-chain B). The construct cloned to obtain this crystal structure expresses a forced heterodimeric fusion protein of BCL6 (WT) and MIZ1 BTB domain sequences connected by a linker peptide. The electron density from the linker peptide is not reported in the final structure, so the PDB coordinates were used in the simulation files preparated without further modifications. The BCL6-PATZ1 heterodimer structure was created using BCL6-BTB monomer (PDB entry 1R29) and PATZ1-BTB monomer (PDB entry 6GUV). The three BCL6 residues mutated to aid the crystallization process (C8Q; C67R; C84N) (Ahmad et al, 2003) were back mutated to WT using the Mutate Residue Plugin of VMD (Humphrey et al, 1996). Missing residues in the A2/B3 loop (75–105) of the PATZ1 structure were homology-modeled as described previously (Piepoli et al, 2020). The LRF-BTB structure (PDB entry 2NN2) was similarly modeled to fill the missing coordinates for A2/B3 residues 66–71 with ModLoop (Fiser & Sali, 2003). The PATZ2/ZBTB24-BTB domain (1–126) was homology-modeled with the PRIMO suite (Hatherley et al, 2016) using BACH1, BACH2, MIZ1, BCL6, and PATZ1 structures as

templates. Similarly, the ThPOK/cKrox/ZBTB15/ZBTB7b-BTB domain (1–144) was homology-modeled using SWISSMODEL (Waterhouse et al, 2018) using LRF/ZBTB7a as a template. All modeled heterodimer structures were generated with the PRISM docking server (Baspinar et al, 2014) by selecting the pose with the highest energy score.

### Molecular dynamics simulations

MD simulations were performed in NAMD using the CHARMM36 force field parameters (Phillips et al, 2005; Best et al, 2012). The simulation environment was prepared in VMD (Humphrey et al, 1996). BTB dimer structures were centered in a solvent box padded with a 10-Å layer of water in every direction. The solvent was modeled using TIP3W water molecules, ionized with 0.15 M KCl. Periodic boundary conditions were applied in which long-range electrostatic interactions were treated using the particle mesh Ewald method (Darden et al, 1999) with a cutoff distance of 12 Å. The structural analysis by molecular simulation includes an initial run of minimization at constant temperature and constant volume (NVT). In the case of BCL6-PATZ1 and LRF-ThPOK heterodimers, the protein dimers were minimized for 30,000 steps. A series of short runs (2 ns) with ramping temperature at 10 K intervals (from 280 to 310 K) was performed to reach the final running temperature of 310 K. All simulations were then performed at a constant temperature of 310 K in isothermal and isobaric conditions (NPT) after minimization, for a total of 500 ns.

### Estimating free energy differences by MM-GBSA calculations

Based on RMSD calculations, we determined a time interval with the most stable conformation of each structure by calculating RMSD values over 500 ns. Continuous RMSD values within 1-Å variation were considered a stable interval used for further analysis. For each stable conformation, a coordinate file (pdb) and a trajectory file (dcd) were saved separately for monomers and for the complex (dimer) without solvent. The MD log file results obtained with NAMD were used to retrieve the energy components used in the molecular mechanics/generalized born surface area (MM-GBSA) calculations (Hou et al, 2011). The free energy of dimerization ($\Delta G$) is estimated by the equation:

$$\Delta G = \Delta E_{int} + \Delta E_{ele} + \Delta G_{sol} + \Delta E_{vdW} \qquad (1)$$

$$\Delta G_{sol} = \Delta G_{sol}^{PB} + \Delta G_{sol}^{SA}$$

where $\Delta E_{int}$ represents the changes in intermolecular interactions calculated using the combined change in bond, angle, dihedral, and improper energies. We note that MM-GBSA does not directly predict the binding free energies, mainly because of the implicit solvent approximation used. Rather, they are compared with experimental $\Delta G$ values inferred from binding constants, via $\Delta G = RT \ln K_d$, via a linear regression; see, for example, Mulakala and Viswanadhan (2013), Adasme-Carreño et al (2014), and Sun et al (2014). Therefore, they should be treated as scoring functions, and their actual values should not be directly converted to $K_d$s. As a result, MM-GBSA values listed in this work are used to assess the relative contributions from its individual terms. $\Delta E_{ele}$ and $\Delta E_{vdW}$ represent the

change in electrostatic and van der Waals energies, respectively. $\Delta G_{sol}$ is the sum of the electrostatic solvation energy (polar contribution); $\Delta G_{sol}^{PB}$ calculated via the Poisson–Boltzmann (PB) approximation and the non-electrostatic solvation component (nonpolar contribution) $\Delta G_{sol}^{SA}$ that is related to the solvent accessibility (SA) of the residues. The Generalized Born implicit solvent (GBIS), based on the Poisson Boltzmann model, calculates the polar contribution, whereas the nonpolar energy is estimated by the SASA. Each energy component term was first extracted separately for the single monomers and for the dimer complex from the MD log files with the pynamd script (Radak, 2021). To calculate each term in the final equation, the sum of the values of the individual monomers was subtracted from the value of the complex. For each frame, the sum of all finalized components was used to calculate the $\Delta G$ of binding using Equation (1). All energy terms were calculated for every frame, and standard error was added to their average. For example, the $\Delta E_{vdW}$ term is $\Delta E_{vdW} = \langle \Delta E_{vdW}^{complex} \rangle - [\langle \Delta E_{vdW}^{monomer1} \rangle + \langle \Delta E_{vdW}^{monomer2} \rangle]$.

## Data Availability

The BTB dimer models generated in this study are available in ModelArchive (modelarchive.org) with the accession codes ma-olypj (PATZ1 homodimer), ma-1iskk (PATZ2 homodimer), ma-zhxm1 (ThPOK homodimer), ma-hf06e (PATZ1-PATZ2 heterodimer), ma-ql2m8 (BCL6-PATZ1 heterodimer), and ma-wrsln (LRF-ThPOK heterodimer). The primary data for imaging experiments have been deposited to BioImage Archive (accession number S-BIAD418).

## Supplementary Information

## Acknowledgements

We thank Dr. Remy Bosselut for helpful comments during the preparation of the manuscript. This work was supported by TÜBİTAK grant number (118Z015 and 20AG007) to B Erman (117F389) to C Atilgan and EMBO Installation Grant (4163) to O Adebali.

### Author Contributions

S Piepoli: conceptualization, data curation, formal analysis, validation, investigation, visualization, methodology, and writing—original draft, review, and editing.
S Barakat: data curation, formal analysis, and investigation.
L Nogay: data curation, formal analysis, and investigation.
B Şimşek: data curation, formal analysis, and investigation.
U Akkose: data curation and software.
H Taskiran: formal analysis, investigation, and methodology.
N Tolay: data curation, formal analysis, and investigation.
M Gezen: formal analysis and investigation.

CY Yeşilada: data curation and investigation.

M Tuncay: formal analysis.

O Adebali: data curation, software, formal analysis, funding acquisition, investigation, and project administration.

C Atilgan: conceptualization, resources, formal analysis, supervision, funding acquisition, project administration, and writing—original draft, review, and editing.

B Erman: conceptualization, resources, data curation, software, formal analysis, supervision, funding acquisition, validation, investigation, visualization, methodology, project administration, and writing—original draft, review, and editing.

## Conflict of Interest Statement

The authors declare that they have no conflict of interest.

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
