## [Reviewer comments · Life Science Alliance]

Life Science Alliance

Sibling rivalry among the ZBTB transcription factor family: homo vs. heterodimers

Sofia Piepoli, Sarah Barakat, Liyne Nogay, Büşra Şimşek, Umit Akkose, Hakan Taskiran, Nazife Tolay, Melike Gezen, Canberk Yeşilada, Mustafa Tuncay, Ogun Adebali, Canan Atilgan, and Batu Erman

DOI: <https://doi.org/10.26508/lsa.202201474>

Corresponding author(s): Batu Erman, Bogazici University and Ogun Adebali, Sabanci University

Review Timeline:

Submission Date:	2022-04-05
Editorial Decision:	2022-04-08
Revision Received:	2022-06-24
Editorial Decision:	2022-07-25
Revision Received:	2022-08-15
Editorial Decision:	2022-08-19
Revision Received:	2022-08-26
Accepted:	2022-08-29

Scientific Editor: Novella Guidi

Transaction Report:

Please note that the manuscript was reviewed at Review Commons and these reports were taken into account in the decision-making process at Life Science Alliance.

April 8, 2022

Re: Life Science Alliance manuscript #LSA-2022-01474

Dr. Batu Erman
Bogazici University
Department of Molecular Biology and Genetics
Istanbul 34543
Turkey

Dear Dr. Erman,

Thank you for submitting your manuscript entitled "Sibling rivalry among the ZBTB transcription factor family: homo vs. heterodimers" to Life Science Alliance. The manuscript was submitted and reviewed via Review Commons. The authors then chose to transfer their somewhat revised manuscript, along with the reviewers' comments and a proposed revised plan to Life Science Alliance (LSA). The reviewer comments and revision plan was assessed at LSA, and LSA editors deemed that the manuscript could be further considered at LSA provided the authors revise the manuscript, in accordance to what they have laid out in the pbp rebuttal / revision plan. We, thus, encourage you to submit a revised manuscript to us that includes all the experiments you have laid out in their Revision plan.

Given that new data will be added to the revised manuscript, the revision will have to be looked at by a set of referees, most likely the same ones as Review Commons.

Thank you for this interesting contribution to Life Science Alliance. We are looking forward to receiving your revised manuscript.

Sincerely,

B. MANUSCRIPT ORGANIZATION AND FORMATTING:

Revision Plan

Manuscript number: RC-2022-01247

Corresponding author(s): Batu, Erman

1. General Statements

The manuscript focuses on the choice between heterodimerization and homodimerization of BTB domains. The human genome encodes 49 ZBTB transcription factors that use this N-terminal domain for dimerization and C-terminal Zinc finger domains for DNA binding. While structural studies identify BTB proteins as obligate homodimers, the structural conservation among BTB domains suggest that heterodimers may in fact be forming *in vivo*. We describe a novel unbiased interaction assay that led to two major findings:

- 1) that contrary to expectations, most BTB domains do not form heterodimers. In this robust assay, these proteins can readily form homodimers.
- 2) that among the assayed partners, PATZ1-PATZ2 is unique in forming heterodimers in this assay.

The manuscript also describes the energetic contributions to BTB domain dimerization by molecular dynamics analyses. We also analyze gene expression profiles of all ZBTB family members to identify the prerequisite conditions for heterodimerization.

BTB homodimers form symmetrical structures, some of which, including PATZ1, BCL6 and PLZF have been reported to interact with symmetrical nuclear corepressor proteins to function as transcriptional repressors. Our demonstration of the presence of a PATZ1-PATZ2 heterodimer now will allow the addressing of the interaction partners of such asymmetrical heterodimers. This will pave the way to the identification of a unique subset of PATZ1-PATZ2 target genes among the already known PATZ1 and PATZ2 gene targets.

While the F2H assay is not a novel technology, our version of it is a highly novel method to assess the choice of homo vs heterodimerization, which may be applied to the investigation of other protein interactions. As BTB domains are not restricted to the ZBTB family, the F2H assay will be useful for numerous investigators studying the function of other proteins containing a BTB domain. The modular BTB domain is used not only by zinc finger transcription factors but also in proteins that control biological processes that regulate the cytoskeleton, multimerization and gating of ion channels and protein ubiquitination/degradation. All of these BTB proteins are candidates for the F2H assay, potentially revealing an unaddressed combinatorial specificity of heterodimerization.

We thank the reviewers for their constructive comments about the manuscript. They have commented that the F2H dataset generated in this study is robust and they have re-iterated our

Revision Plan

viewpoint that the network of interactions between BTB domain containing proteins is of general interest.

2. Description of the planned revisions

To address comments by Reviewer#1, we plan to conduct the following additional experiments:

- In response to point 6, we plan to conduct a co-expression and co-immunoprecipitation experiment with epitope tagged-PATZ1 and PATZ2 BTB domain proteins.

3. Description of the revisions that have already been incorporated in the transferred manuscript

Reference style has been changed to the Harvard style.

Following the suggestions by Reviewer #1, we modified our manuscript addressing the following points:

Point 1. We think our study contributes to the literature in two significant ways. Firstly, despite structural and sequence conservation, and expression overlaps, BTB domains of ZBTB protein do not readily form heterodimers. This is a striking observation that cannot be predicted from current structural knowledge. Secondly, the PATZ1-PATZ2 interaction represents a clear exception to the rule. We document the formation of this heterodimer which has biological implications for the function of these transcription factors. To emphasize the striking absence of heterodimers, we changed the section heading from "PATZ1-PATZ2 is a unique BTB domain heterodimer" to "Despite significant structural similarity, BTB domains prefer to form homodimers over heterodimers."

Point 2. The reviewer requested BACH2-PATZ1 interaction data using the F2H assay. We have conducted these experiments and found that there is no interaction between BACH2 and PATZ1 BTB domains. The lack of a heterodimeric interaction between these two proteins in the F2H assay indicates that the BTB domains are not sufficient to reproduce an interaction between the full-length proteins. In fact, the Y2H assay that led to the identification of this interaction, (Kobayashi et al. 2000), is inconclusive about the domain requirements for the interaction. In response to point 2 of Reviewer#1, we plan to conduct additional F2H experiments with NAC1-BTB domain in the future. However, this manuscript focuses only on ZBTB protein and NAC1, like BACH2, does not contain a Zinc finger DNA-binding domain and lies outside the scope of the current study.

The prey fished out in this Y2H assay, conducted with the BACH2-BTB domain as bait, was in fact a PATZ1 (MAZR) clone that contained not only the BTB domain, but also the AT-hook motif and 7 ZN-Fingers in the C-terminus. Moreover, Table1 of this referenced paper demonstrates

Revision Plan

that while PATZ1-BTB domain can efficiently homodimerize (144 fold), it is inconclusive whether it can heterodimerize with the BTB domain of BACH2. Thus, we think that the BACH2-PATZ1 interaction requires sequences outside the BTB domains. This interpretation is consistent with our findings in our F2H assay. As BACH2 is a BTB-bZip factor, we do not think including this negative result in this manuscript which focuses on ZBTB family members is critical. However, we are willing to add this data as supplemental information if desired.

Point 3. Figure 2D was replaced to show better superposition of GFP and RFP signals for MIZ1. Moreover, a spreadsheet is provided in the supplementary data 1 ("1.F2H_quantification.xlsx") that summarizes for each dimer, the number of images acquired, the number of cells per image that scored positive for a GFP focus and the number of co-localizing RFP foci scored as an interaction (1) or not (0). The number of interactions reported in Figure 2E were calculated from this spreadsheet. The primary data for imaging experiments have been deposited to BiImage Archive (accession number TMP_1648242736382).

Point 4. MIZ1-BCL6 heterodimer has been documented in the literature, as we discuss in the manuscript. These were covalently linked, forced heterodimers, which we did not observe in our F2H assay. MIZ1 clearly can form homodimers in this assay, indicating that N-terminal fusion constructs do not interfere with dimerization. Because of these findings, we think that novel experiments using C-terminal GFP-tagged BTB domains will not significantly change the conclusions of our F2H assay and prefer to reserve these experiments to future studies. Also, crystal structures of BTB domains show an obligate dimer where the N-terminus of one monomer is in close proximity to the C-terminus of the second monomer. This observation indicates that a putative steric hindrance arising from an N-terminal GFP fusion in the F2H assay would also likely be observed if the fusion protein was placed in the C-terminus. Therefore, we do not think that this kind of experiment will significantly change the interaction specificity of the dimers.

Point 5. The reviewer questioned the efficiency of recruitment to LacO loci in the F2H assay. The experimental approach for using the LacI-GFP nanobody (GBP) to recruit GFP-tagged proteins to the LacO locus is well characterized (Herce et al., 2013 and Tang et al. 2013). We included these references in the revised manuscript.

Moreover, we added a spreadsheet as supplementary data 2 ("2.BTB GFPexpression.xlsx") where we analyze the number of cells with GFP expression that also contain a green focus, indicative of the recruitment of the tag-GFP BTB domain fusion protein to the LacO sequences through the function of the LacI-GBP protein. We observe that the efficiency of colocalization is directly correlated with the ability to form GFP-foci. There may be unique and unknown properties of BTB domains (such as interactions with host cell factors) that influence nuclear localization. We recognize that this may be an important finding about BTB domain function, but it is not the focus of the current study. Although different BTB domains may result in different efficiency of recruitment to the LacO focus, we focused our analysis on RFP-GFP colocalization only on GFP-foci containing cells (Figure 2E), we scored these interactions as positive.

Revision Plan

While the F2H assay is not a novel technology, our version of it is a highly novel method to assess the choice of homo vs heterodimerization which may be applied to the investigation of other protein interactions.

Point 6. Please see planned revisions.

Point 7. We agree with Reviewer#1 and to mention the exception represented by MIZ1 lack of β 1-strand we added one sentence to the manuscript, in the first paragraph of "Results" section.

Minor comments.

1. There is only 1 amino acid difference in the BTB domain protein sequence of human and mouse PATZ1 and we specified this in the methods. The difference is at position 91 (within the A2/B3 loop). We do not think this residue has an influence on dimerization as it is outside of the interaction interface which we previously determined in our X-ray crystallography studies (Piepoli et al., 2020).

2. The specific GFP-nanobody (VHHdomain cAbGFP4 from Camelus dromedarius) protein sequence is the following:

MADVQLVESGGALVQPGGSLRLSCAASGFPVNRYSMRWYRQAPGKEREWVAGMSSAGDRS
SYEDSVKGRFTISRDDARNTVYLQMNSLKPEDTAVYYCNVNVGFYWGQGTQVTVSS

PDB Structure reference: 3OGO, (Kubala et al., 2010). We added this information to the "Methods" section.

Following the suggestions by Reviewer #2, we modified our manuscript addressing the following points:

Point 1. We agree with the reviewer that the absence of heterodimerization among BTB domains is a striking observation. To address this point, we changed the section heading from "PATZ1-PATZ2 is a unique BTB domain heterodimer" to "Despite significant structural similarity, BTB domains prefer to form homodimers over heterodimers."

Point 2. We added a detailed discussion of the MM-GBSA approach in the Methods section including three new references.

Minor comments

Point 3. We wanted to point out the existence of an unknown subset of genes that could be regulated by the heterodimerization of PATZ1 and PATZ2 rather than ignoring the known gene targets of PATZ1 and PATZ2 homodimers. The text now reads: "We deduce that, besides the genes regulated by PATZ1-PATZ1 or PATZ2-PATZ2 homodimers, a further subset of target genes is likely regulated by PATZ1-PATZ2 heterodimers."

Point 4. In the F2H assay 64 pairs of BTB domain proteins were actually evaluated in 64 distinct experiments (8 homodimers and 56 heterodimers). In fact, for each protein, two plasmids were constructed, one containing the GFP-tag, one the RFP-tag. We thank the reviewer for the chance to clarify this point, but nothing was changed in the text.

Revision Plan

Point 5. “ND” in Figure 2B indicates that no cell image (0%) for the tested BTB pair was found to contain an overlap of green and red foci (co-localization). We added a spreadsheet in the supplementary data 1 (“1.F2H_quantification.xlsx”). Please see the discussion of Reviewer#1, point 3.

Point 6. We presume the reviewer is interested in heterodimers rather than homodimers in this point. For heterodimers both RFP/GFP and GFP/RFP pairs were tested. Figure 2B does not show an average but rather the percentage of all positive interactions (green and red foci overlap) over the total number of cell images. For the only heterodimer observed (PATZ1-PATZ2), both GFP-PATZ1 RFP-PATZ2 and GFP-PATZ2 RFP-PATZ1 pairs scored positive. The all-cell count for each BTB pair is reported in Figure 2E and includes results from both RFP/GFP and GFP/RFP tested pairs. nothing was changed in the text.

Point 7. We changed “in vitro” to “cell-based” in the “Abstract”.

Point 8. In figure S1 we report the correlation values (R) and p-values calculated for the selected ZBTB gene pairs in each cell type individually. We now added a folder with the original data for R and p-values calculated for each gene pair in the ZBTB family for each cell type (Supplementary data “3.ZBTB_co-expression_correlation.zip”) as an extension of figure S1.

In figure S2, the correlation data from all cell types (with either significant or not significant p-values) were aggregated to calculate an overall correlation of expression between gene pairs. Specific information that can be highlighted when considering individual cell type (Figure S1 and extension) is lost when we use all cell types in figure S2 to get the overall picture of general co-expression patterns.

Point 9. Labels in Figure S2 and S6 use the common gene names also used in the manuscript. We define both names for the indicated gene/protein in the “Introduction” section. In the “Discussion” section at paragraph 4, we added a description of the cluster of highly correlated genes formed by 22 genes of the ZBTB family extracted from the ImmGen RNA-expression database. The clustering of the expression profile of these genes (Zbtb2, Zbtb24 (Patz2), Zfp161 (Zbtb14), Zbtb25, Zbtb48 (Hkr3), Zbtb17 (Miz1), Zbtb40, Zbtb49, Mynn (Zbtb31), Zbtb33 (Kaiso), Zbtb1, Zbtb43, Zbtb26, Zbtb44, Zbtb11, Zbtb37, Zbtb8b, Hic2 (Zbtb30), Zbtb4, Zbtb38, Zbtb8a and Zbtb20) suggests their possible co-regulation which is a pre-requisite for interaction.

Green highlight has been corrected to indicate the Bcl6-Miz1 pair in Figure S2.

We have not yet tested the ZBTB9-PATZ1 pair in the F2H assay. The focus of this work is on ZBTB family members with structural information. There is no crystallographic information on ZBTB9, which is why we excluded it from our analysis for this manuscript. In future work we intend to complete the F2H matrix but at the current time we think this lies outside of the scope of our manuscript.

Revision Plan

Point 10. We have performed a novel analysis of the ZBTB phylogenetic tree and provided a new Figure S6 with highlighted bootstrap values considered significant (>0.5). We also describe the novel approach in the Methods section. The tree does not significantly change from the previous version.

Point 11. We stand by our playful choice of manuscript title and think that it will attract readers' interest, but we leave it up to the editor of the journal whether it conforms to its standards.

Following the suggestions by Reviewer #3, we modified our manuscript addressing the following points for the minor comments:

Point 1. A "time interval" was chosen for each simulation by considering the variation of the RMSD plot. Continuous RMSD values within 1 Å variation were considered a stable interval used for further analysis. We added this detail in the "Methods" section.

Point 2. Wording order has been changed to "Bcl6-Patz1" to fit the more frequently used one. All-upper-case style refer to protein names, while capitalizing only the first letter refers to gene names, conventionally.

Point 3. Homodimers and heterodimers analyzed have now been listed in the manuscript.

Point 4. The number of salt bridges strongly contribute to the value of ΔE_{ele} . An approximate count of salt bridges for each pair has been added to table 2 and table 3.

Point 5. We cannot understand this minor comment and we do not think a change is warranted.

Point 6. BSA is correlated to ΔG but only partly contributes to final energy score (ΔG). For some cases the electrostatic component is the major component (PATZ1). We have made a clearer discussion of these effects in the revised version.

7. The RMSD value between the equilibrated conformation found through our MD simulation and the crystal structure is calculated as 1.4 Å at 150 ns and 1.1 Å at 400 ns of the simulation. This shows reliability of our model over the deposited crystal structure and this info was added to the text.

8. We changed the description of the tables in question in the "Methods" section as follows: "All energy terms were calculated for every frame and standard error was added to their average."

Revision Plan

4. Description of analyses that authors prefer not to carry out

In response to point 2 of Reviewer#1, we plan to conduct additional F2H experiments with NAC1-BTB domain in the future. However, this manuscript focuses only on ZBTB protein and NAC1, like BACH2, does not contain a Zinc finger DNA-binding domain and lies outside the scope of the current study.

In response to point 4 of Reviewer#1, MIZ1-BCL6 heterodimer has been documented in the literature, as we discuss in the manuscript. These were covalently linked, forced heterodimers, which we did not observe in our F2H assay. MIZ1 clearly can form homodimers in this assay, indicating that N-terminal fusion constructs do not interfere with dimerization. Because of these findings, we think that novel experiments using C-terminal GFP-tagged BTB domains will not significantly change the conclusions of our F2H assay and prefer to reserve these experiments to future studies.

In response to the minor comment 9 of Reviewer#2, we have not yet tested the ZBTB9-PATZ1 pair in the F2H assay. The focus of this work is on ZBTB family members with structural information. There is no crystallographic information on ZBTB9, which is why we excluded it from our analysis for this manuscript. In future work we intend to complete the F2H matrix but at the current time we think this lies outside of the scope of our manuscript.

July 25, 2022

Re: Life Science Alliance manuscript #LSA-2022-01474R

Dr. Batu Erman
Bogazici University
Department of Molecular Biology and Genetics
Istanbul 34543
Turkey

Dear Dr. Erman,

Thank you for submitting your revised manuscript entitled "Sibling rivalry among the ZBTB transcription factor family: homo vs. heterodimers" to Life Science Alliance. The manuscript has been seen by the original reviewers whose comments are appended below. While the reviewers continue to be overall positive about the work in terms of its suitability for Life Science Alliance, some important issues remain. Please address the remaining Reviewer 1' concerns.

Our general policy is that papers are considered through only one revision cycle; however, given that the suggested changes are relatively minor, we are open to one additional short round of revision. Please note that I will expect to make a final decision without additional reviewer input upon resubmission.

Please submit the final revision within one month, along with a letter that includes a point by point response to the remaining reviewer comments.

To upload the revised version of your manuscript, please log in to your account: <https://lsa.msubmit.net/cgi-bin/main.plex>
You will be guided to complete the submission of your revised manuscript and to fill in all necessary information.

B. MANUSCRIPT ORGANIZATION AND FORMATTING:

Sincerely,

Reviewer #1 (Comments to the Authors (Required)):

Reviewer 1 general comments on the revised manuscript:

BTB-domains mediate the homodimerisation of over 50 mammalian transcription factors (TFs). BTB-directed heterodimerisation of TFs has been reported, though is less well characterized. Using a fluorescent two-hybrid (F2H) assay in transiently-transfected mammalian cells, the authors studied heterodimerisation between eight specific BTB domains. Of these, only PATZ1 and PATZ2 could form heterodimers in the assay. Using published gene expression profiles, it was found that the genes encoding PATZ1 and PATZ2 are co-expressed in immune cells. The authors then constructed models of BTB-domain heterodimers, and used molecular dynamics simulations study the energetic determinants of homo vs heterodimerisation.

In their revised manuscript, the authors have carried out additional experimentation (Figure 2f) to convincingly show an interaction between the BTB domains of PATZ1 and PATZ2 in transfected mammalian cells. However, I do not consider that the F2H assay is sufficiently refined - for example, the homodimeric interaction of the MIZ1 BTB domain (Figures 2c and S1b) is not convincing, and importantly it cannot be concluded that the BTB domains of MIZ1 and BCL6 do not interact in the F2H assay.

Reviewer 1 comments on the authors' response:

Revision Plan

2. Description of the planned revisions

To address comments by Reviewer#1, we plan to conduct the following additional experiments: - In response to point 6, we plan to conduct a co-expression and co-immunoprecipitation experiment with epitope tagged-PATZ1 and PATZ2 BTB domain proteins.

Reviewer 1's comment:

This additional experiment (Figure 2f) convincingly demonstrates an interaction between PATZ1 and PATZ2.

Following the suggestions by Reviewer #1, we modified our manuscript addressing the following points:

Point 1. We think our study contributes to the literature in two significant ways. Firstly, despite structural and sequence conservation, and expression overlaps, BTB domains of ZBTB protein do not readily form heterodimers. This is a striking observation that cannot be predicted from current structural knowledge. Secondly, the PATZ1-PATZ2 interaction represents a clear exception to the rule. We document the formation of this heterodimer which has biological implications for the function of these transcription factors. To emphasize the striking absence of heterodimers, we changed the section heading from "PATZ1-PATZ2 is a unique BTB domain heterodimer" to "Despite significant structural similarity, BTB domains prefer to form homodimers over heterodimers."

Reviewer 1's comment:

This is change in heading is fine, though I still consider that not all of the "negative" interactions are convincing (see comments under points 3 and 4 below).

Point 2. The reviewer requested BACH2-PATZ1 interaction data using the F2H assay. We have conducted these experiments and found that there is no interaction between BACH2 and PATZ1 BTB domains. The lack of a heterodimeric interaction between these two proteins in the F2H assay indicates that the BTB domains are not sufficient to reproduce an interaction between the full-length proteins. In fact, the Y2H assay that led to the identification of this interaction, (Kobayashi et al. 2000), is inconclusive about the domain requirements for the interaction. In response to point 2 of Reviewer#1, we plan to conduct additional F2H experiments with NAC1-BTB domain in the future. However, this manuscript focuses only on ZBTB protein and NAC1, like BACH2, does not contain a Zinc finger DNA-binding domain and lies outside the scope of the current study. The prey fished out in this Y2H assay, conducted with the BACH2-BTB domain as bait, was in fact a PATZ1 (MAZR) clone that contained not only the BTB domain, but also the AT-hook motif and 7 ZN-Fingers in the C-terminus. Moreover, Table1 of this referenced paper demonstrates that while PATZ1-BTB domain can efficiently homodimerize (144 fold), it is inconclusive whether it can heterodimerize with the BTB domain of BACH2. Thus, we think that the BACH2-PATZ1 interaction requires sequences outside the BTB domains. This interpretation is consistent with our findings in our F2H assay. As BACH2 is a BTB-bZip factor, we do not think including this negative result in this manuscript which focuses on ZBTB family members is critical. However, we are willing to add this data as supplemental information if desired.

Reviewer 1's comment:

Please add this information into the supplementary material.

Point 3. Figure 2D was replaced to show better superposition of GFP and RFP signals for MIZ1. Moreover, a spreadsheet is provided in the supplementary data 1 ("1.F2H_quantification.xlsx") that summarizes for each dimer, the number of images acquired, the number of cells per image that scored positive for a GFP focus and the number of co-localizing RFP foci scored as an interaction (1) or not (0). The number of interactions reported in Figure 2E were calculated from this spreadsheet. The primary data for imaging experiments have been deposited to BiImage Archive (accession number TMP_1648242736382).

Reviewer 1's comment:

The "Miz1 panel" of Figure 2C (Figure 2D in the original manuscript) appears to be the same as in the original manuscript and is not convincing. Please show many more examples of your positive MIZ1 interactions in the supplementary information (show examples of cells that are both "positive" and "negative"- the .xlsx file suggests there were 263 cells showing positive interaction). For all your interactions, please also show a superposition of the two channels - this would clarify the presentation

(i.e. as in Figure 2e). (note - I could not find reference to the .xlsx file in the text, so please add if missing).

Point 4. MIZ1-BCL6 heterodimer has been documented in the literature, as we discuss in the manuscript. These were covalently linked, forced heterodimers, which we did not observe in our F2H assay. MIZ1 clearly can form homodimers in this assay, indicating that N-terminal fusion constructs do not interfere with dimerization. Because of these findings, we think that novel experiments using C-terminal GFP-tagged BTB domains will not significantly change the conclusions of our F2H assay and prefer to reserve these experiments to future studies. Also, crystal structures of BTB domains show an obligate dimer where the N-terminus of one monomer is in close proximity to the C-terminus of the second monomer. This observation indicates that a putative steric hindrance arising from an N-terminal GFP fusion in the F2H assay would also likely be observed if the fusion protein was placed in the C-terminus. Therefore, we do not think that this kind of experiment will significantly change the interaction specificity of the dimers.

Reviewer 1's comment:

This response is confusing. The authors have now in fact conducted some experiments in which the GFP and RFP tags were placed on the C-termini of the BTB proteins (revised lines 318 - 338 and revised Figure S1). Some of the homodimeric interactions are not at all convincing in these experiments (between PATZ1-GFP and RFP-PATZ1 panel c; between MIZ1-GFP and RFP-MIZ1). Judging from comparison of the panels shown in Figure 2C and Figure S1, it appears that the interaction between GFP-PATZ1 with RFP-PATZ1 is stronger than that between PATZ1-GFP and RFP-PATZ1, suggesting that the proximity of the tags may indeed be hindering the interaction in the latter situation.

Although most BTB proteins (including PATZ1 and BCL6) are domain-swapped dimers in which the N-terminus of one domain is close to the C-terminus of the other, this region is not close to the main BTB dimerization interface (i.e. the "central" region of the dimer that contains alpha1) so the proximity of the tags in a BTB-GFP/RFP-BTB situation might not be detrimental to the interaction (especially as the positioning of alpha6 at the C-terminus of the BTB domain is often somewhat mobile - this might account for differences in efficiency of homodimeric interaction between the various BTB domains tested in Figure S1 panels a,b,c,d and e). In contrast, MIZ1 lacks a beta1 strand, so any tag placed on the N-terminus of the MIZ1 BTB domain would be jammed against the main dimerization interface and would be expected to have a greater effect on both homo- and heterodimerisation.

Point 5. The reviewer questioned the efficiency of recruitment to LacO loci in the F2H assay. The experimental approach for using the LacI-GFP nanobody (GBP) to recruit GFP-tagged proteins to the LacO locus is well characterized (Herce et al., 2013 and Tang et al. 2013). We included these references in the revised manuscript.

Moreover, we added a spreadsheet as supplementary data 2 ("2.BTB GFPexpression.xlsx") where we analyze the number of cells with GFP expression that also contain a green focus, indicative of the recruitment of the tag-GFP BTB domain fusion protein to the LacO sequences through the function of the LacI-GBP protein. We observe that the efficiency of colocalization is directly correlated with the ability to form GFP-foci. There may be unique and unknown properties of BTB domains (such as interactions with host cell factors) that influence nuclear localization. We recognize that this may be an important finding about BTB domain function, but it is not the focus of the current study. Although different BTB domains may result in different efficiency of recruitment to the LacO focus, we focused our analysis on RFP-GFP colocalization only on GFP-foci containing cells (Figure 2E), we scored these interactions as positive.

Reviewer 1's comment:

Thank you for clarifying. (note - I could not find reference to the .xlsx file in the text, so please add if missing).

Revision Plan

While the F2H assay is not a novel technology, our version of it is a highly novel method to assess the choice of homo vs heterodimerization which may be applied to the investigation of other protein interactions.

Point 6. Please see planned revisions.

Reviewer 1's comment:

Now OK, see above.

Point 7. We agree with Reviewer#1 and to mention the exception represented by MIZ1 lack of β 1-strand we added one sentence to the manuscript, in the first paragraph of "Results" section.

Reviewer 1's comment:

Now OK.

Minor comments.

1. There is only 1 amino acid difference in the BTB domain protein sequence of human and mouse PATZ1 and we specified this in the methods. The difference is at position 91 (within the A2/B3 loop). We do not think this residue has an influence on dimerization as it is outside of the interaction interface which we previously determined in our X-ray crystallography studies (Piepoli et al., 2020).

Reviewer 1's comment:

OK, thank you for clarifying.

2. The specific GFP-nanobody (VHHdomain cAbGFP4 from Camelus dromedarius) protein sequence is the following:
MADVQLVESGGALVQPGGSLRLSCAASGFPVNRYSMRWYRQAPGKEREWVAGMSSAGDRS
SYEDSVKGRFTISRDDARNTVYLQMNSLKPEDTAVYYCNVNVGFEYWGQGTQVTVSS
PDB Structure reference: 3OGO, (Kubala et al., 2010). We added this information to the "Methods" section.
Reviewer 1's comment:
Now OK.

Reviewer #3 (Comments to the Authors (Required)):

The focus on heterodimerization of the BTB domain is of great importance in considering protein-protein interactions involving the BTB domain.

The authors have previously shown that the BTB domains of PATZ1 and PATZ2 heterodimerize in cell biological and molecular dynamics analyses.

They supported this point with the additional experiment in the revise.

(minor point)

In Table 1.

Denotations "Patz1", "Patz2", "Bcl6" and "Miz1" should be replaced by "PATZ1", "PATZ2", "BCL6" and "MIZ1".

We thank you for giving us the opportunity to address Reviewer 1's questions.

We understand that Reviewer 1 primarily asked for the following extra data:

1. More examples of MIZ1BTB domain homodimerizing and also some examples of failed interactions,
2. More examples of MIZ1BTB-BCL6BTB domains not heterodimerizing
3. Demonstration of the inability of BACH2-BTB and PATZ1-BTB domains heterodimer formation.

To address these points, we have now made Figure S1 and shifted the remaining supplemental figures. Fig S1 contains:

(a) Five panels of positive MIZ1-BTB-MIZ1-BTB interactions, (b) five panels of negative MIZ1-BTB-MIZ1-BTB interactions, (c) five panels of negative MIZ1-BTB-BCL6-BTB (GFP-RFP) interactions, (d) five panels of negative MIZ1-BTB-BCL6-BTB (RFP-GFP), (e) two panels of positive BACH2-BTB and BACH2-BTB (GFP-RFP) interactions and (f) two panels of negative PATZ1-BTB and BACH2-BTB (GFP-RFP) interactions. We tried to fit as many examples of cells that do or do not show interactions in the F2H assay as possible. Moreover, as requested, in the methods section, we refer to table S2 which has a quantification of these observations, as requested.

In these new figures we have also provided a panel showing the superposition of the two channels, as requested. The manuscript file remains unchanged with the exception of minor typographical corrections and a discussion of our new Fig S1, highlighted in yellow.

Our responses to the reviewer's final points:

Point 1: More data provided in Fig. S1. as discussed above.

Point 2: Discussion added to legend of Fig. S1.

Point 3: More data provided in Fig. S1. as discussed above.

Point 4: We apologize for the confusion. We agree with the reviewer that fusing the fluorescent protein to the C-terminus of one BTB domain and assessing whether there is an interaction with an N-terminally tagged BTB domain is not appropriate. In each of the crystal structures of BTB domains reported in the literature, the N-terminus of one monomer is very close to the C-terminus of the second monomer. Thus it would be expected to have steric hindrance if bulky fluorescent proteins are fused to these ends. This is why in our original construction of the F2H assay, we fused the two fluorescent proteins (GFP and RFP) to the N-termini of both monomers, which are distant from each other in the crystal structures. We mentioned this fact in the rebuttal to the original Review Commons review but conducted these experiments anyway as one of the reviewers asked for them. In fact in Fig. S1, as the reviewer and we expect, we report that there is no homodimeric interaction between PATZ1 monomers, LRF monomers, BCL6 monomers, MIZ1 monomers, when the GFP and RFP are placed close to each other. Surprisingly the PATZ1-PATZ2 heterodimer forms even when the GFP and RFP are placed on the N- and C-Termini respectively. We discuss this in the manuscript.

Reviewer 1 General Point: "I do not consider that the F2H assay is sufficiently refined - for example, the homodimeric interaction of the MIZ1 BTB domain (Figures 2c and S1b) is not convincing, and importantly it cannot be concluded that the BTB domains of MIZ1 and BCL6 do not interact in the F2H assay. "

We hope that the new Fig. S1 which is more convincing about the results of the F2H assay. Moreover, we note that all of the primary data for imaging experiments have been deposited to Biolmage Archive with accession number S-BIAD418 (<https://www.ebi.ac.uk/biostudies/BiolImages/studies/S-BIAD418?query=S-BIAD418>). We had mentioned this in the original revision plan and it is referenced in the Data Availability section.

We would again like to draw your attention to the graphical abstract of our study which we generated using data from the PDB of all the components of our novel F2H assay. This figure shows a model of the interacting GFP, GBP, LacI and BTB domains that are either shown as monomers, homodimers or heterodimers. All protein structures are displayed to scale, connected by modelled linker regions, where necessary. The figure shows a tetramer of the LacI DNA binding domain in cyan and green, each monomer fused to a GBP in grey. Each GBP binds to a GFP molecule (in green) which is fused to a monomer of a BTB domain (in blue, purple or yellow). BTB domain structures were derived from the crystal structures of PATZ1 and BCL-6 BTB homodimers and a homology modelled PATZ2 BTB domain. The four scenarios of interactions observed in

the F2H assay are shown from left to right; a monomer, a homodimer with two GFP tags, a homodimer with a GFP and an RFP tag and a heterodimer with a GFP and an RFP tag. If our manuscript is accepted for publication, we would be delighted to propose this figure as a cover image for the issue.

August 19, 2022

RE: Life Science Alliance Manuscript #LSA-2022-01474RR

Dr. Batu Erman
Bogazici University
Department of Molecular Biology and Genetics
Istanbul 34543
Turkey

Dear Dr. Erman,

Thank you for submitting your revised manuscript entitled "Sibling rivalry among the ZBTB transcription factor family: homo vs. heterodimers". We would be happy to publish your paper in Life Science Alliance pending final revisions necessary to meet our formatting guidelines.

- please consult our manuscript preparation guidelines <https://www.life-science-alliance.org/manuscript-prep> and make sure your manuscript sections are in the correct order
- please add an abstract, summary blurb, category and the Twitter handle of your host institute/organization as well as your own or/and one of the authors in our system
- please use the [10 author names, et al.] format in your references (i.e. limit the author names to the first 10)
- on page 9, please indicate to which figure the figure callout for the panels c-d belong
- Thank you for providing us with a Graphical Abstract/Cover Art; LSA does not consider this a Graphical Abstract, since a Graphical Abstract is meant to be a self-explanatory visual summary of the main findings of the article that should capture the content of the article at a single glance. We will, however, include this as a potential option for when we select a cover image for this issue.

Figure Check:

- please upload your main figures as single files; these will be displayed in-line in the HTML version of your paper, so please provide them as single page files (Figure 2 currently spans 2 pages); we do not have a limit on the number of main figures and these can be split if necessary for space
- Figure 2C needs scale bars
- please add sizes next to all blots

A. FINAL FILES:

-- Summary blurb (enter in submission system): A short text summarizing in a single sentence the study (max. 200 characters including spaces). This text is used in conjunction with the titles of papers, hence should be informative and complementary to the title. It should describe the context and significance of the findings for a general readership; it should be written in the

present tense and refer to the work in the third person. Author names should not be mentioned.

B. MANUSCRIPT ORGANIZATION AND FORMATTING:

Sincerely,

August 29, 2022

RE: Life Science Alliance Manuscript #LSA-2022-01474RRR

Dr. Batu Erman
Bogazici University
Department of Molecular Biology and Genetics
Istanbul 34543
Turkey

Dear Dr. Erman,

Thank you for submitting your Research Article entitled "Sibling rivalry among the ZBTB transcription factor family: homo vs. heterodimers". It is a pleasure to let you know that your manuscript is now accepted for publication in Life Science Alliance. Congratulations on this interesting work.

DISTRIBUTION OF MATERIALS:

Again, congratulations on a very nice paper. I hope you found the review process to be constructive and are pleased with how the manuscript was handled editorially. We look forward to future exciting submissions from your lab.

Sincerely,
